# Central role of nitric oxide in ozone production in the upper tropical troposphere over the Atlantic Ocean and West Africa

Ivan Tadic[1], Clara M. Nussbaumer[1], Birger Bohn[2], Hartwig Harder[1], Daniel Marno[1], Monica Martinez[1], Florian Obersteiner[3], Uwe Parchatka[1], Andrea Pozzer[1,4], Roland Rohloff[1], Martin Zöger[5], Jos Lelieveld[1,6] and Horst Fischer[1]

[1]Atmospheric Chemistry Department, Max Planck Institute for Chemistry, Mainz, Germany
[2]Institute of Energy and Climate Research, IEK-8: Troposphere, Forschungszentrum Jülich GmbH, Jülich, Germany
[3]Karlsruhe Institute of Technology, Karlsruhe, Germany
[4]Earth System Physics section, The Abdus Salam International Centre for Theoretical Physics, Trieste, Italy
[5]Flight Experiments, German Aerospace Center (DLR), Oberpfaffenhofen, Germany
[6]Climate and Atmosphere Research Center, The Cyprus Institute, Nicosia, Cyprus

**Correspondence:** Ivan Tadic (i.tadic@mpic.de) or Horst Fischer (horst.fischer@mpic.de)

**Abstract.** Mechanisms of tropospheric ozone ($O_3$) formation are generally well understood. However, studies reporting on net ozone production rates (NOPRs) directly derived from in situ observations are challenging, and are sparse in number. To analyze the role of nitric oxide (NO) in net ozone production in the upper tropical troposphere above the Atlantic Ocean and the West African continent, we present in situ trace gas observations obtained during the CAFE-Africa (**C**hemistry of the **A**tmosphere: **F**ield **E**xperiment in **Africa**) campaign in August and September 2018. The vertical profile of in situ measured NO along the flight tracks reveals lowest NO mixing ratios of less than 20 pptv between 2 and 8 km altitude and highest mixing ratios of 0.15-0.2 ppbv above 12 km altitude. Spatial distribution of tropospheric NO above 12 km altitude shows that the sporadically enhanced local mixing ratios (> 0.4 ppbv) occur over the West African continent, which we attribute to episodic lightning events. Measured $O_3$ shows little variability in mixing ratios at 60-70 ppbv, with slightly decreasing and increasing tendencies towards the boundary layer and stratosphere, respectively. Concurrent measurements of CO, $CH_4$, OH and $HO_2$ and $H_2O$ enable calculations of NOPRs along the flight tracks and reveal net ozone destruction at -0.6 to -0.2 ppbv $h^{-1}$ below 6 km altitude and balance of production and destruction around 7-8 km altitude. We report vertical average NOPRs of 0.2-0.4 ppbv $h^{-1}$ above 12 km altitude with NOPRs occasionally larger than 0.5 ppbv $h^{-1}$ over West Africa coincident with enhanced NO. We compare the observational results to simulated data retrieved from the general circulation ECHAM/MESSy Atmospheric Chemistry (EMAC) model. Although the comparison of mean vertical profiles of NO and $O_3$ indicates good agreement, local deviations between measured and modelled NO are substantial. The vertical tendencies in NOPRs calculated from simulated data largely reproduce those from in situ experimental data. However, the simulation results do not agree well with NOPRs over the West African continent. Both measurements and simulations indicate that ozone formation in the upper tropical troposphere is $NO_x$-limited.

# 1 Introduction

The importance of nitrogen oxides ($NO_x$ = NO + $NO_2$) and ozone ($O_3$) in the photochemistry of the atmosphere is widely acknowledged. Both NO and $NO_2$ are toxic gases, which degrade surface air quality and regulate the abundance of secondary tropospheric oxidants (Hosaynali Beygi et al., 2011; Silvern et al., 2018). They are the propagating agents in the formation of $O_3$ and govern photochemical ozone production and removal from the atmosphere (Bozem et al., 2017; Schroeder et al., 2017). Ozone is a greenhouse gas, negatively affects human health and causes ecosystem damage (Jaffe et al., 2018). It is the primary precursor of the hydroxyl (OH) radical, which determines the oxidation capacity of the atmosphere and directly controls the concentrations of methane ($CH_4$), carbon monoxide (CO) and many volatile organic compounds (VOCs) (Thornton et al., 2002; Bozem et al., 2017). The U.S. Clean Air Act identified ozone as a criteria air pollutant in the 1970s (Jaffe et al., 2018). Since then and especially in the last decades, increasing effort has been put in the understanding and mitigation of tropospheric ozone pollution (Fiore et al., 2002; Dentener et al., 2005; West and Fiore, 2005; Lelieveld et al., 2009, Pusede et al., 2015; Jaffe et al., 2018; Nussbaumer and Cohen, 2020; Tadic et al., 2020). To further resolve the complexity of scientific and policy-related issues of the $NO_x$-$O_3$-VOCs relationship, careful evaluation of model simulations against in situ measurement data is required (Sillman et al., 1995).

Photochemical ozone formation in the troposphere has been comprehensively described in the literature. Briefly, $O_3$ is photochemically formed in chemical reactions between $NO_x$, $HO_x$ (= OH + $HO_2$) and VOCs (Crutzen, 1974, Schroeder et al., 2017). VOCs are here referred to as RH where R stands for an organic residual. The oxidation of CO, $CH_4$ and VOCs by OH results in the production of $HO_2$ and peroxy radicals ($RO_2$).

$$OH + CO + O_2 \rightarrow HO_2 + CO_2 \tag{R1}$$
$$OH + CH_4 + O_2 \rightarrow CH_3O_2 + H_2O \tag{R2}$$
$$OH + RH + O_2 \rightarrow RO_2 + H_2O \tag{R3}$$

$HO_2$ and $RO_2$ (including $CH_3O_2$ and further organic peroxy radicals) rapidly oxidize NO to $NO_2$, which will yield $O_3$ in its subsequent photolysis (reaction R6) followed by recombination of atomic ground-state oxygen with molecular oxygen (reaction R7) (Thornton et al., 2002).

$$NO + HO_2 \rightarrow NO_2 + OH \tag{R4}$$
$$NO + RO_2 \rightarrow NO_2 + RO \tag{R5}$$
$$NO_2 + h\nu \rightarrow NO + O(^3P) \tag{R6}$$
$$O(^3P) + O_2 + M \rightarrow O_3 + M \tag{R7}$$

The net effect of reaction R1-R7 on $HO_x$ and $NO_x$ is zero, which is why both act as catalysts in photochemical $O_3$ production. Ozone loss is due to photolysis (and subsequent reaction of $O(^1D)$ with $H_2O$) and reactions of $O_3$ with OH and $HO_2$.

$$O_3 + h\nu \rightarrow O(^1D) + O_2 \tag{R8}$$

$$O(^1D) + H_2O \rightarrow 2OH \tag{R9}$$

$$O_3 + OH \rightarrow HO_2 + O_2 \tag{R10}$$

$$O_3 + HO_2 \rightarrow OH + 2O_2 \tag{R11}$$

Note that the deactivation of $O(^1D)$ to $O(^3P)$ via collisions with $N_2$ and $O_2$ will result in the reformation of $O_3$ (Bozem et al., 2017; Tadic et al., 2020). We express the portion of $O_3$ that is effectively lost via photolysis by $\alpha$ (see section 2.2). In this study, we neglect chemical loss reactions of $O_3$ with alkenes, sulphides and halogen radicals. Note that reactions R8-R11 will

be referred to as gross ozone loss, while the rate-limiting reactions of NO with $HO_2$ or $RO_2$ to produce $NO_2$ will be referred to as gross ozone production (Zanis et al., 2000a; Thornton et al., 2002). The difference between these two quantities will yield net ozone production, conventionally given in units of ppbv h$^{-1}$ (Bozem et al., 2017) or ppbv d$^{-1}$ (Tadic et al., 2020).

The dependency of NOPRs on ambient levels of $NO_x$ is highly non-linear (Bozem et al., 2017). Due to the above-mentioned chemistry gross ozone loss will naturally prevail over gross ozone production at low $NO_x$. Increasing ambient $NO_x$ will result

in a linear increase in ozone formation such that the chemical air mass will shift from net destruction to net production in ozone (Bozem et al., 2017; Schroeder et al., 2017). However, at a certain NO mixing ratio, which depends on ambient levels of $HO_x$ and VOCs, adding more NO to the system will result in a saturation in ozone formation and eventually in a decrease in net ozone production towards higher NO levels (Tadic et al., 2020). This is due to the reaction of $NO_2$ with OH to produce $HNO_3$ followed by its deposition to the surface.

$$NO_2 + OH + M \rightarrow HNO_3 + M \tag{R12}$$

Reaction R12 will decrease the pool of available $HO_x$ and $NO_x$ radicals from the atmosphere to produce $O_3$ (Thornton et al., 2002). Ozone formation hence crucially depends on whether $NO_x$ or VOCs are available in excess. These two atmospheric states are commonly referred to as either VOC-limited (if $NO_x$ is available in excess) or as $NO_x$-limited (if VOCs are available in excess) (Sillman et al., 1995; Sillman et al., 2003; Duncan et al., 2010; Nussbaumer and Cohen, 2020; Tadic et al., 2020).

The lifetime of $NO_x$ in the atmosphere varies from a few hours in the planetary boundary layer (PBL) to 1-2 weeks in the upper troposphere/lower stratosphere (UTLS) (Beirle et al., 2010). In the latter, the reaction of $NO_2$ with OH during daytime and $NO_3$ formation at nighttime is slowed down due to low ambient pressure and temperature. Transport of $NO_x$ from polluted regions to pristine areas is limited due to the short lifetime of $NO_x$ in the PBL (Reed et al., 2016), which is why $NO_x$ in the troposphere can vary over several orders of magnitude (Miyazaki et al., 2017; Tadic et al., 2020). Whilst measurements

performed in remote and pristine regions, such as in the unpolluted South Atlantic marine boundary layer (MBL), have reported $NO_x$ mixing ratios of only a few tens of pptv (Hosaynali Beygi et al., 2011; Fischer et al., 2015), $NO_x$ mixing ratios in urban areas can exceed several tens of ppbv (Lu et al., 2010). Measurements obtained in the polluted MBL around the Arabian Peninsula have shown that $NO_x$ mixing ratios can locally exceed several tens of ppbv even in marine environments in the proximity to strong emission sources such as passing ships or downwind of megacities (Tadic et al., 2020).

Ground-level $NO_x$ emissions include fossil fuel combustion, biomass burning and soil emissions (Silvern et al., 2018). Lightning $NO_x$ ($LNO_x$), aircraft emissions, and, to a lesser extent, convective uplift of potentially $NO_x$-rich planetary boundary air and intrusion of stratospheric air are predominant sources of $NO_x$ in the upper troposphere (Bozem et al., 2017; Miyazaki et al., 2017). However, estimates of lightning produced $NO_x$ are uncertain (Beirle et al., 2010; Miyazaki et al., 2017) and can have large implications on the photochemistry of the upper troposphere such as over tropical areas where lightning flash rates are enhanced (Christian et al., 2003; Tost et al. 2007).

A number of previous studies have performed measurements in the region of interest, the troposphere over the Atlantic Ocean and the West Africa (Lelieveld et al., 2004; Aghedo et al., 2007; Saunois et al., 2009; Real et al., 2010; Bourgeois et al., 2020). Lelieveld et al. (2004) indicated that positive ozone trends in the marine boundary layer over the Atlantic are likely caused by an increase in anthropogenic emissions of nitrogen oxides. Aghedo et al. (2007) showed that lightning acts as a major source of tropospheric $NO_x$, leading to a significant increase in middle and upper tropospheric ozone over the African continent. Saunois et al. (2009) described results from airborne measurements in the region during the AMMA project. Deploying a two-dimensional model for further analysis, Saunois et al. determined positive trends in photochemical net ozone production in the boundary layer over West Africa. There are also results from the ATom airborne mission, which measured vertical profiles of $O_3$ in the troposphere over the Atlantic Ocean (Bourgeois et al., 2020), which we will use to validate the results presented here. Real et al. (2010) investigated downwind $O_3$ production in pollution plumes in the mid and upper troposphere and determined mean net ozone production rates of 2.6 ppbv/day over a period of 10 days. However, studies reporting on vertical profiles and spatial distributions of nitric oxide, ozone and net ozone production rates as part of one coherent measurement project in the troposphere over the West African continent and the Atlantic Ocean are absent.

In the present study, we characterize the distribution of NO and the role of NO in photochemical processes in the upper tropical troposphere above the Atlantic Ocean and West Africa. The structure of the paper is as follows: we provide methodological, practical and technical information about the campaign and deployed instrumentation in Sect. 2. In Sect. 3 we present in situ NO and $O_3$ data obtained during the campaign including vertical profiles and spatial distributions. Based on concurrent measurements of CO, $CH_4$, OH and $HO_2$, $H_2O$, the actinic flux density, pressure and temperature net $O_3$ production rates (NOPRs) were calculated along the flight tracks. We also provide a comparison of the observational results to simulated data retrieved from the 3-D EMAC model and analyze the dependency of NOPRs on ambient NO. In Sect. 4, we summarize our results and draw conclusions based on our findings.

## 2 Experimental

### 2.1 CAFE-Africa campaign

The airborne measurement-based CAFE-Africa project took place in August and September 2018 in the tropical troposphere over the Central Atlantic Ocean and the West African continent. Starting from and returning to the international airport on Sal, Cape Verde (16.75° N, 22.95° W) a total of 14 scientific measurement flights (MFs) was carried out with the German High

**A**ltitude and **Lo**ng-range research aircraft (HALO). For the analysis of the MFs, we consecutively numerate each MF, starting with MF03 on August 07, 2018 for the ferry flight from Oberpfaffenhofen (Germany, Deutsches Zentrum für Luft- und Raumfahrt) to Sal (Cape Verde Islands) on and ending with MF16 on September 07, 2018 for the back ferry flight from Sal to Oberpfaffenhofen on. The test flights MF01 and MF02 conducted over Germany are not included in this study. MF03-MF16 covered a latitudinal range from 8° S to 48.2° N and a longitudinal range from 47.9° W to 12.5° E and reached maximum flight altitudes of about 15 km. Before landing at the home base airport in Sal, a fixed-altitude leg of 30 min duration at FL150 (~4,600 m altitude) was flown for calibration purposes. Take off (T/O) time of the flights was typically 9 or 10 UTC, except for MF08 with T/O at 4 UTC and landing around 13 UTC and MF11 with T/O at 16 UTC and landing around 1 UTC the next day.

The location of the campaign home base on Sal provided the unique possibility to analyze the impact of the **I**nter-**T**ropical **c**onvergence **z**one (ITCZ) on physical and chemical processes in the airspace above the Atlantic Ocean and the West African continent. The ITCZ is a low-pressure region evolving near the equator, which is characterized by deep convection, strong precipitation and frequent lightning (Collier and Hughes, 2011), producing nitrogen oxides, mostly as NO through the Zeldovich reactions from atmospheric $N_2$ and $O_2$. The campaign was performed in late summer (August and September) 2018 when the ITCZ had reached its northernmost position at around 5-15° N (Collier and Hughes, 2011) and was henceforth located only a few degrees in latitude to the south of the campaign base at 16.75° N. The flight tracks of the 14 MFs performed during the campaign are shown in Fig. 1. An overview of the corresponding flight dates and objectives of each particular MF is given in the supplementary Table ST1.

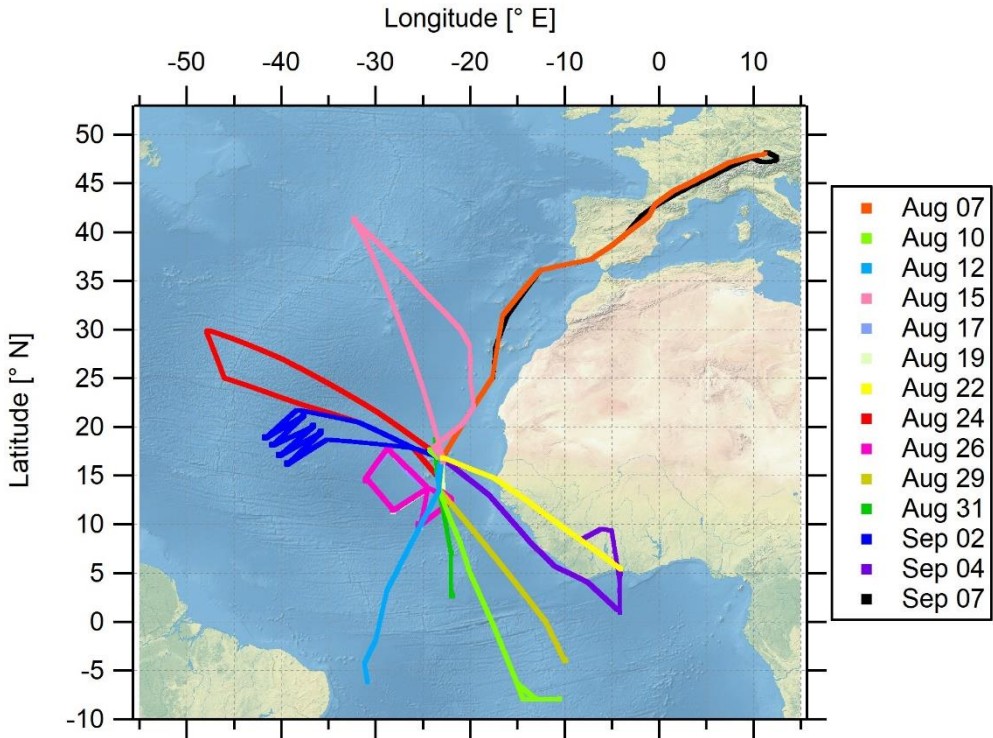

**Figure 1: Spatial orientation of the measurement flight tracks of during CAFE-Africa. Note that MF07 (August 17, 2018), MF08 (August 19, 2018) and MF11 (August 26, 2018) had identical flight tracks.**

## 2.2 Chemiluminescent detection of NO

In situ measurements of $NO_x$ on ground-based and mobile platforms are challenging in terms of the demand for high sensitivity and high precision (Tadic et al., 2020). During CAFE-Africa, we deployed a modified commercially available chemiluminescent detector *CLD 790 SR* (ECO Physics Inc., Dürnten, Switzerland) on-board HALO. It is the same instrument that has been used during previous shipborne (Hosaynali Beygi et al., 2011; Tadic et al., 2020) and airborne campaigns (Bozem et al., 2017). The measurement method is based on the gas phase reaction of NO with $O_3$, which will partly produce excited-state $NO_2^*$ followed by the spontaneous emission (chemiluminescence) of a photon (Ridley and Howlett, 1967; Ryerson et al., 2000).

$$NO + O_3 \rightarrow NO_2^* + O_2 \tag{R13}$$

$$NO_2^* \rightarrow NO_2 + h\nu \ (\lambda > 600 \text{ nm}) \tag{R14}$$

Photons generated through the emissions from excited-state $NO_2^*$, which are directly proportional to the NO concentration in the sample flow (Ridley and Howlett, 1967), are detected by a photomultiplier tube and converted to an electric pulse. Carrying out the oxidation of NO by $O_3$ at low-pressure (7-8 mbar) and in a temperature-stabilized (25 °C) main reaction chamber,

minimizes quenching (non-radiative de-excitation of $NO_2^*$ via collisions) (Reed et al., 2016; Tadic et al., 2020). Detector dark noise and artefacts due to the reaction of $O_3$ with species other than NO (such as alkenes and sulphides) is corrected for by using a pre-chamber setup, as first described by Ridley and Howlett (1967). A residual instrumental background (due to memory effects within the instrument) is corrected for by regularly sampling synthetic zero air (Tadic et al., 2020). During the MFs, we sampled zero air from a tank (17 l composite tank, AVOX) with a Purafil-activated carbon adsorbent installed downstream of the zero-air tank to ensure NO-free zero-air measurements. The residual instrumental background of the NO measurement was calculated at 5 pptv from measurements obtained at nighttime during MF11. As chemiluminescent detection of NO is an indirect measurement method, regular calibrations against a known standard are needed. During the MFs we diluted the secondary NO standard ($1.187 \pm 0.036$ ppmv NO in $N_2$) at a mass flow of 8.6 sccm in a zero-air flow of 3.44 SLM (standard litre per minute) resulting in NO calibration gas mixing ratios of ~3 ppbv. NO calibrations measurements were performed six to eight times during a MF of 9-10 h duration by manually initiating calibration slots consisting of 2 min zero-air measurement, 2 min NO calibration and 2 min zero-air measurement, similar to previous deployments of the instrument (see Tadic et al., 2020).

The limit of detection (LOD) of the NO data was calculated at 9 pptv from the FWHM (full width at half maximum) of a Gauss Fit applied to the distribution of 1 s NO data obtained at nighttime during MF11 (see supplement Figure S1). Analogously we estimate the LOD of the NO data at 1 min time resolution to be 5 pptv from the FWHM of a Gauss Fit applied to the distribution of 1 min NO data obtained at nighttime during MF11. The precision of the NO data was calculated from the average reproducibility of all in-flight calibrations to be 5 % at $1\sigma$. The uncertainty in the used secondary standard mixing ratio was 3 %. The total measurement uncertainty (TMU) of the NO data was estimated at 6 % as the quadratic sum of the precision and the uncertainty of the secondary standard (Tadic et al., 2020).

$$\text{TMU(NO)} = \sqrt{(5\ \%)^2 + (3\ \%)^2} \approx 6\ \% \tag{1}$$

## 2.3 Further measurements used in this study

$O_3$ was quantified with a chemiluminescence detector calibrated by a UV photometer (**F**ast **AIR**borne **O**zone Instrument; Zahn et al., 2012). CO and $CH_4$ were measured by mid-infrared quantum cascade laser absorption spectroscopy (QCLAS) with TRISTAR, a multi-channel spectrometer (Schiller et al., 2008; Tadic et al., 2017). OH and $HO_2$ radicals were measured by laser-induced fluorescence with the custom-built HORUS instrument (Marno et al., 2020). Note that both OH and $HO_2$ data are preliminary. We conservatively estimate the total relative measurement uncertainty of the OH and $HO_2$ data at 50 %. Spectrally resolved actinic flux density measurements were obtained with two spectro-radiometers (upward- and downward-looking) installed on the top and bottom of the aircraft fuselage, respectively. The particular photolysis frequencies were calculated from the actinic flux density spectra between 280 and 650 nm (Bohn and Lohse, 2017). The uncertainty in the used $j$-values was estimated to be 13 %. Water vapor mixing ratio and further derived humidity parameters were measured by SHARC (**S**ophisticated **H**ygrometer for **A**tmospheric **R**esear**C**h) based on dual path direct absorption measurement by a

tunable diode laser (TDL) system (Krautstrunk and Giez, 2012). The measurement range of SHARC covers the whole
troposphere and lower stratosphere (5-40000 ppmv) with an absolute accuracy of 5 % (+1 ppmv). The BAMAHAS (**BA**sic
**HA**LO **M**easurement **A**nd **S**ensor System) provided measurements of temperature and pressure (Krautstrunk and Giez, 2012).
All instruments deployed on the aircraft have been developed to meet the high standards of airborne measurements in terms
of operability, accuracy and sensitivity. Table 1 lists the used instrumentation with the associated total measurement
uncertainties. A reference is given regarding the use of each method during previous measurements.

**Table 1: List of performed observations with the corresponding total measurement uncertainty (given as a percentage) during
CAFE-Africa. The last column provides a reference regarding the practical use of the used measurement/instrument.**

| Measurement | Technique / method | TMU | reference |
|---|---|---|---|
| NO | chemiluminescence | 6 % | Tadic et al., 2020 |
| $O_3$ | UV photometry / chemiluminescence | 2.5 % | Zahn et al., 2012 |
| CO | QCLAS | 4.3 % | Tadic et al., 2017 |
| $CH_4$ | QCLAS | 0.3 % | Schiller et al., 2008 |
| OH | LIF | 50 % | Marno et al., 2020 |
| $HO_2$ | LIF with chemical conversion | 50 % | Marno et al., 2020 |
| $H_2O$ | TDL | 5 % | Krautstrunk and Giez, 2012 |
| $j(O^1D)$ | spectral radiometer | 13 % | Bohn and Lohse, 2017 |

**2.4 ECHAM/MESSy Atmospheric Chemistry (EMAC) model and data analysis**

EMAC is a 3-D global general circulation, atmospheric chemistry-climate model, which has been used and described in a
number of previous studies (Roeckner et al., 2006; Jöckel et al., 2010; Sander et al., 2019; Tadic et al., 2020). Briefly, EMAC
comprises the 5th generation of the **E**uropean **C**enter **Ham**burg (ECHAM5) circulation model (Roeckner et al., 2006) and the
**M**odular **E**arth **S**ubmodel **Sy**stem (MESSy) in the version 2.52 (Jöckel et al., 2010). Here we use the model in the T63L47
resolution with a spatial resolution of roughly 1.8° × 1.8°, 47 vertical levels and one data point every 6 min. The model has
been weakly nudged towards the ECMWF ERA-Interim data (Jeucke et al., 1996; Berrisford et al., 2009). The chemical
mechanism (the Mainz Organic Mechanism, MOM) and the photolysis rate calculations used in this work have been presented
in Sander et al. (2019) and in Sander et al. (2014), respectively. The Emissions Database for Global Atmospheric Research
(EDGARv4.3.2, Crippa et al. 2018) were used for anthropogenic emissions, while biomass burning emissions were from the
GFAS (Global Fire Assimilation System) database with a daily temporal resolution (Kaiser et al 2012). Important for this
work, the $NO_x$ emissions from lightning activity have been estimated using the submodel LNOX (Tost et al., 2007), where the
parameterization by Grewe et al. (2001) was applied. The global $NO_x$ emissions from lightning were scaled to 6.3 Tg(N)/yr,
following Miyazaki et al. (2014). Tracer and aerosol wet and dry deposition were estimated following Tost et al. (2006) and
Kerkweg et al. (2006), respectively. The $NO_x$ soil biogenic emission flux is calculated based on a semi-empirical emission
algorithm implementation by Yienger and Levy II (1995; Kerkweg et al., 2006). For the current study we use EMAC
simulations of NO, $O_3$, OH, $HO_2$, $CH_3O_2$, specific humidity, $j(O^1D)$, temperature and pressure spatially interpolated to the
flight tracks (latitude, longitude and altitude). Based on the simulations we perform calculations of $\alpha$ and NOPRs along the

flight tracks (see section 2.5). To synchronize the time stamp of the model data (6 min) with the measurement data (1 min), we have calculated a running mean of the measurement data within $\pm 2$ min around the simulated data point along the measurement flight tracks such that every sixth measurement data point (if available) was neglected.

## 2.5 Calculation of net ozone production rates (NOPRs)

Calculation of NOPRs utilizes the chemical reactions related to ozone formation described in the introduction. EMAC model
calculations show that during CAFE-Africa $CH_3O_2$ represents on average 80 % of the sum of all organic peroxy radicals with respect to ozone formation at typical flight altitudes of 200 hPa (and even up to 90 % at lower altitudes). Model calculations further show that the sum of $HO_2$ and $CH_3O_2$ represents on average 95 % of the sum of $HO_2$ and all organic peroxy radicals ($RO_2$) yielding that the ratio $(HO_2+CH_3O_2)/(HO_2+RO_2)$ is practically one. In analogy to Tadic et al. (2017), we estimated $RO_2$ as the sum of all organic peroxy radicals with less than four carbon atoms. See supplementary Table ST2 for an overview of
the used organic peroxy radicals. Therewith we calculate photochemical gross production of $O_3$ by the rate-limiting reaction of NO with $HO_2$ and $CH_3O_2$ (Thornton et al., 2002; Bozem et al., 2017).

$$P(O_3) = [NO] \cdot (k_{NO+HO_2}[HO_2] + k_{NO+CH_3O_2}[CH_3O_2]). \tag{2}$$

The IUPAC Task Group on Atmospheric Chemical Kinetic Data Evaluation (Atkinson et al. 2004; Atkinson et al., 2006) provides rate coefficients used in this study. Note that other studies use $P(O_3)$gross as an acronym for $P(O_3)$ in Eq. 2. The
photochemical lifetimes of both $HO_2$ and $CH_3O_2$ are similar with respect to self-reactions yielding hydrogen peroxide and methyl hydroperoxide and reactions with NO and $HO_x$ (Bozem et al., 2017). We further assume photostationary steady state (PSS) for the probed air masses. As the typical time to acquire PSS during CAFE-Africa varied between 40 s at 2 km altitude and about 70-80 s at 15 km altitude (Mannschreck et al., 2004; Tadic et al., 2020), we can calculate the concentration of $CH_3O_2$ by the equation derived by Bozem et al. (2017).

$$[CH_3O_2] = \frac{k_{OH+CH_4}[CH_4]}{k_{OH+CO}[CO]} \cdot [HO_2] \tag{3}$$

Note that the reaction of CO with OH represents the dominant term in $HO_2$ production during CAFE-Africa. Assuming mixing ratios of 500 ppbv and 100 pptv for $H_2$ and HCHO, respectively, we find that $HO_2$ production rate from the reaction of OH with CO is on average 5 times greater than the sum of the $HO_2$ production rates from photolysis of HCHO and the reactions of HCHO and $H_2$ with OH during CAFE-Africa. Note that the assumed mixing ratio of 100 pptv represents a rather
conservative upper estimate for HCHO in the upper troposphere. As mentioned above, ozone loss due to photolysis (and subsequent reaction of $O(^1D)$ with $H_2O$) will only partly lead to a net loss effect as most $O(^1D)$ will deactivate via collisions with air molecules, mostly $N_2$ and $O_2$, to $O(^3P)$ and reform $O_3$ in the subsequent reaction with $O_2$. The share of $O_3$ photolysis that will eventually lead to a net loss in $O_3$, can be calculated using Eq. 4 (Bozem et al., 2017; Tadic et al., 2020).

$$\alpha = \frac{k_{O(^1D)+H_2O}[H_2O]}{k_{O(^1D)+H_2O}[H_2O]+k_{O(^1D)+N_2}[N_2]+k_{O(^1D)+O_2}[O_2]} \tag{4}$$

In the troposphere, $\alpha$ ranges from about 15 % in the PBL to 1 % in the upper troposphere, where absolute humidity is very low (Bozem et al., 2017). Further loss processes of $O_3$ (reactions with alkenes, sulphides and halogen radicals) are considered small and are therefore neglected in this study. Equation 5 then gives gross loss of ozone.

$$L(O_3) = [O_3] \cdot (\alpha \cdot j(O^1D) + k_{OH+O_3}[OH] + k_{HO_2+O_3}[HO_2]) \qquad (5)$$

$j(O^1D)$ expresses the photolysis frequency of ozone to $O(^1D)$. The NOPR is given as the difference of the gross ozone
production rate (Eq. 2) and the gross ozone loss rate (Eq. 5) (Lin et al., 1988, Cantrell et al., 2003).

$$NOPR = P(O_3) - L(O_3) =$$

$$[NO] \cdot (k_{NO+HO_2}[HO_2] + k_{NO+CH_3O_2}[CH_3O_2]) - [O_3] \cdot (\alpha \cdot j(O^1D) + k_{OH+O_3}[OH] + k_{HO_2+O_3}[HO_2]) \qquad (6)$$

Note that other studies use $P(O_3)$net as an acronym for NOPR in Eq. 6.

### 3 Results

**3.1 Vertical profiles of NO and O₃ in the tropical troposphere**

In the following, we will investigate averages of the vertical profiles, which are calculated based on an altitude bin width of 1 km. The profiles are calculated with respect to the centre of the particular bin, e.g. the average at 3.5 km includes all data points obtained at or above 3 km altitude and below 4 km altitude. Data are filtered for stratospheric influence by removing all data points for which concurrent $O_3$ is larger than 100 ppbv; a conservative criterion which has been discussed by Prather et al.
(2011). Fig. 2 shows the vertical NO and $O_3$ profiles obtained during CAFE-Africa. The orange and blue lines represent vertical average profiles of experimental and model simulated data, respectively. The blue and orange shading in the respective colors represent the $\pm 1$ standard deviation of the vertical averages.

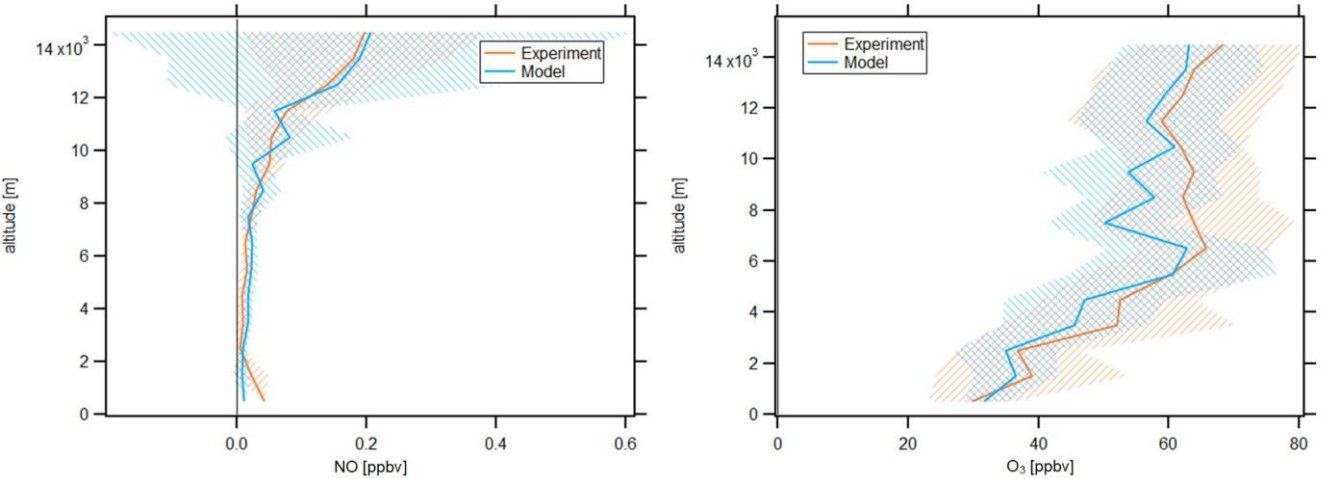

**Figure 2: Vertical NO (left) and O₃ profiles (right) of measured and modelled data along the flight tracks during CAFE-Africa. Note**
**the large variability of simulated NO mixing ratios above 10 km. The figures have been filtered for stratospheric measurements by removing data points for which O₃ exceeds 100 ppbv.**

The vertical profile of measured NO data shows lowest NO mixing ratios of less than 20 pptv observed between 2 and 8 km altitude, which reflect the absence of emission and transport sources at these altitudes. Highest NO mixing ratios of 0.15-0.2 ppbv are observed above 12 km altitude and reflect the increasing amount of lightning-produced $NO_x$, and to a lesser extent influence of relatively $NO_x$-rich stratospheric air. Below 2 km altitude, the vertical profile shows a weak increase of NO, which reproduces the low amount of anthropogenic sources in the investigated MBL and PBL in the proximity to the Cape Verde Islands. This suggests that the contribution of local convective uplift of PBL air to the increased $NO_x$ above 12 km altitude is negligible.

The vertical average profile of simulated NO data is in good agreement with the vertical profile of measured NO data, which is also indicated by the median NO(model)/NO(measurement) ratio throughout the whole campaign at 0.97. Although the vertical profiles are in overall agreement, the variability when comparing single measurement and simulation data points is substantial, as indicated by the large variability of simulated NO data above 10 km altitude. The 25th percentile, 75th percentile and average of the NO(model)/NO(measurement) ratio throughout the whole campaign are 0.25, 2.2 and 2.27, respectively, which illustrate the significant spread among measurement and model data in a comparison of individual data points. The minimum and maximum mixing ratios of modelled NO are zero and 2.13 ppbv, respectively. The minimum and maximum mixing ratios of observed NO are zero and 0.95 ppbv, respectively.

The average vertical profile of measured $O_3$ shows lowest mixing ratios of 30-40 ppbv below 3 km altitude and a steady increase in $O_3$ at about 5-10 ppbv per km altitude to mixing ratios of 60-65 ppbv at 6 km. Above this altitude, $O_3$ is relatively constant (60-65 ppbv) until it further increases above 12 km altitude, simultaneous with the increase in NO.

Although simulated $O_3$ slightly underestimates the measurement data throughout the troposphere, we find that the vertical profiles of simulated and measured $O_3$ data are in again in good agreement. EMAC $O_3$ data match lowest mixing ratios of 30-40 ppbv observed below altitudes of 3 km as well as the vertical gradient in $O_3$ mixing ratios between 3 and 6 km with mixing ratios of 60 ppbv at 6 km altitude. Above 6 km altitude, both the vertically constant $O_3$ mixing ratio as well as the further increase in $O_3$ above 12 km altitude deduced from the measurements are well reproduced by the model. Except for lowest altitudes (< 1 km), it seems that simulated $O_3$ mixing ratios slightly underestimate the measurement data. This is confirmed by the median and average $O_3$(model)/$O_3$(measurement) ratio throughout the campaign at 0.97 and 0.98, respectively, confirming the general agreement between measurements and simulations, as well as the slight underestimation by the latter. The 25th percentile and 75th percentile of the $O_3$(model)/$O_3$(measurement) ratio are 0.85 and 1.11, respectively, and indicate that the spread among single data points when comparing measurement and model for $O_3$ is less than for NO.

$O_3$ profiles observed in this study are in good agreement with results from the ATom mission (Bourgeois et al., 2020). For the June-August season, Bourgeois et al. show that in the tropical troposphere $O_3$ increased with altitude to 50 ppbv at 5-6 km whereas above 9 km $O_3$ varied from 40 to 80 ppbv, supporting the results presented here (see Figures 9 and 10 in Bourgeois et al., 2020).

**3.2 Spatial distribution of NO and O₃ in the upper tropical troposphere**

As most of the measurement time ($> 60\,\%$) of the CAFE-Africa campaign was dedicated to upper tropospheric measurements above 12 km altitude and as both NO and $O_3$ show highest mean mixing ratios above 12 km altitude, we characterize the spatial distribution of NO and $O_3$ above that altitude in the following. Data above 12 km altitude have been aggregated and averaged over a spatial $2° \times 2°$ grid. We again remove stratospheric measurement data by only considering those for which $O_3$ was below 100 ppbv. Note that this does not necessarily exclude influence of mixing with air of stratospheric origin. Figure 3

shows the color-coded spatial NO distributions based on the measured data (left plot) and simulated data (middle plot). The right plot shows the tropospheric average spatial distribution of the point-by-point NO(model)/NO(measurement) ratio above 12 km. Note that the color scales presented in the following emphasize the most relevant features of the spatial distribution. Thus a few single data points might exceed the given color range, such as in the case of simulated NO (Figure 3, middle plot) with single maximum NO mixing ratios of $> 1$ ppbv.

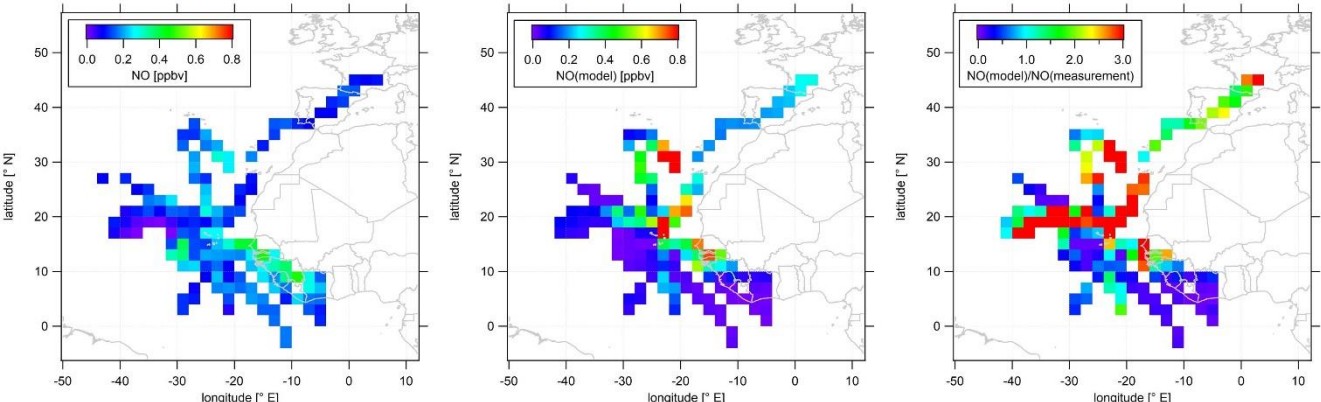


**Figure 3: Color-coded spatial, tropospheric NO distributions above 12 km during CAFE-Africa. The left and middle graphs show the measured and simulated NO, respectively. The right graph shows the spatial distribution of the NO(model)/NO(measurement) ratio. The figures have been filtered for stratospheric measurements by removing data points for which O₃ exceeds 100 ppbv.**

The spatial distribution of NO from the measurement data shows that NO in the upper tropical troposphere above West Africa

and the Atlantic Ocean is generally 0.1-0.2 ppbv. The range in mixing ratios includes lowest NO of less than 20 pptv observed between –30 to –40° E and 16 to 20° N close to a deep convective system without lightning to mixing ratios of 0.1-0.2 ppbv over wide areas over the Atlantic Ocean to (more than) 0.4 ppbv over the West African continent. Although deep convective systems over oceanic regions rarely evolve lightning (Zipser, 1994), we encountered large amounts of NO close to a marine cumulonimbus cloud system with potential lightning activity resulting in more than 0.3 ppbv NO at –28 to –32° E and at 12

to 16° N. The coincidence of the ITCZ (5 to 15° N during August and September) and enhancements in upper tropospheric NO above West Africa underlines the substantial influence of the seasonal migration of the ITCZ and its impact on lightning and nitrogen oxides in the upper tropical troposphere (Zipser, 1994; Xu and Zipser, 2012). The data also suggest a longitudinal increase in NO from about 0.1 ppbv observed at –40° E westbound to 0.4 ppbv above the West African continent. Note that we observe a slight decrease in upper tropospheric NO over the Ivory Coast and partly also over Guinea compared to upper

tropospheric NO over Senegal, although the lightning flash rate over the Ivory Coast and Guinea is reported to be a factor of about 3 larger than that over Senegal (Collier and Hughes, 2011). A linear fit applied to the longitudinal average profile of all NO data weighted by the standard deviation and collected in the troposphere above 12 km altitude between –42° E and –8° E reveals an increase in average NO of 4-5 pptv per degree longitude (see supplement Figure S2).

Although the comparison of the vertical profiles of measured and simulated NO suggests generally good correspondence
between measurements and model simulations, the agreement with respect to the spatial NO distributions in the upper troposphere is much less satisfactory. The EMAC model does not reproduce the large NO enhancements in the area of the ITCZ, as shown in the latitudinal profile of measured and simulated tropospheric NO data above 12 km in Figure S3 in the supplement. On contrary the model tends to underestimate the observations south of 10° N. Interestingly, this holds also for observations above the West African continent (except for the airspace over Senegal around 12 to 14° N) where simulated NO
mixing ratios are highest. This general underestimation of the measurements by the model over large parts of West Africa extends to large parts of the Atlantic Ocean between 5 to 15° N. On the other hand, further north the model tends to overestimate the measurements across large areas north of 16° N and west of –20° E with spatially averaged mixing ratios that exceed 1 ppbv. However, reasonable agreement between measurements and numerical results is observed towards and over Southern Europe.

Figure 4 shows color-coded spatial O$_3$ distributions based on observations (left plot) and simulated data (middle plot). The right plot of Figure 4 shows the average tropospheric distribution of the point-by-point O$_3$(model)/O$_3$(measurement) ratio. In analogy to supplement Figure S3, supplement Figure S4 shows latitudinal profiles of measured and simulated O$_3$ mixing ratios obtained above 12 km in the troposphere.

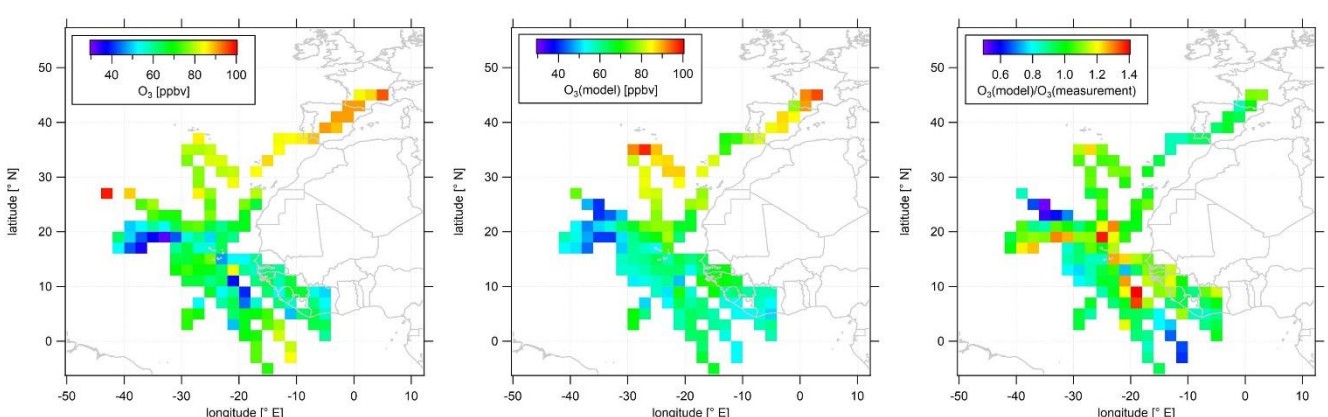

**Figure 4: Color-coded spatial, tropospheric O$_3$ distributions above 12 km during CAFE-Africa. The left and middle graphs show the spatial distribution of measured and simulated O$_3$, respectively. The right graph shows the spatial distribution of the O$_3$(model)/O$_3$(measurement) ratio. The figures have been filtered for stratospheric measurements by removing data points for which O$_3$ exceeds 100 ppbv.**

Measured $O_3$ shows a rather uniform distribution in the upper troposphere above the Atlantic Ocean and West African
continent. The observed mixing ratios range from less than 40 ppbv between –30 to –40° E and 16 to 20° N, areas where NO
is likewise decreased, compared to other regions with more than 80 ppbv towards and over Southern Europe, which partly
reflects the increasing stratospheric impact above 12 km altitude. Over the West African continent, we observed average $O_3$ at
50-70 ppbv, which is in approximate agreement with previous studies (Galanter et al., 2000). Note that measured $O_3$ mixing
ratios over the African continent are not significantly different from $O_3$ mixing ratios over adjacent oceanic areas.

In general, the simulated $O_3$ reproduces the observed absolute $O_3$ mixing ratios in the upper tropical troposphere, as well as
regional tendencies. It is of note that, although the model underestimates NO over the tropical continental area of Africa, $O_3$
is reproduced remarkably well. Nevertheless, the model is not able to reproduce local $O_3$ variations such as at -20° E and 10°
N or at -10° E and 0° N. Moreover, simulated $O_3$ seems to be rather uniformly distributed throughout the whole ITCZ region.
The right panel of Figure 4 further illustrates that the majority of the simulated data points deviate by less than 10-15 % from
the observational data and that larger deviations between model simulations and measurement are mainly restricted to
situations when the measurements show either the lowest or highest mixing ratios, not reproduced by the simulations. The
overall spatial agreement between $O_3$ observation and $O_3$ simulation is also demonstrated in the latitudinal profile given in
supplement Figure S4. Furthermore, supplementary Figures S5 and S6 show 2-D latitudinal/altitudinal distributions of
measured, tropospheric NO and $O_3$, respectively.

**3.3 Net ozone production rates in the tropical troposphere**

In the following, NOPRs are calculated based on Eq. 6 and analyzed both vertically and spatially. The left graph of Figure 5
shows the vertical profile of NOPRs derived from measured and simulated data in orange and blue ($\pm 1$ standard deviation of
the corresponding vertical average), respectively. The middle and right graph show the vertical average profiles of the
components of gross ozone loss and gross ozone production derived from experimental in situ data and simulated data,
respectively. We provide a vertical profile of $\alpha$ calculated based on Eq. 4, for which we obtain good agreement between
measurements and simulations, for which we refer to the left graph of Figure S7 in the supplement. Supplementary Figure S7
also provides a comparison of vertical profiles of measured and simulated $H_2O$ mixing ratios. The vertical profiles are
calculated based on an altitudinal bin width of 1 km and are filtered for stratospheric influence by removing data points for
which $O_3$ is higher than 100 ppbv. Supplement Figure S8 presents latitudinal profiles of NOPRs above 12 km altitude in the
troposphere derived from measured and simulated data. A spatial distribution of OH and $HO_2$ (derived from both measured
and simulated data) is given in the supplement Figure S9.

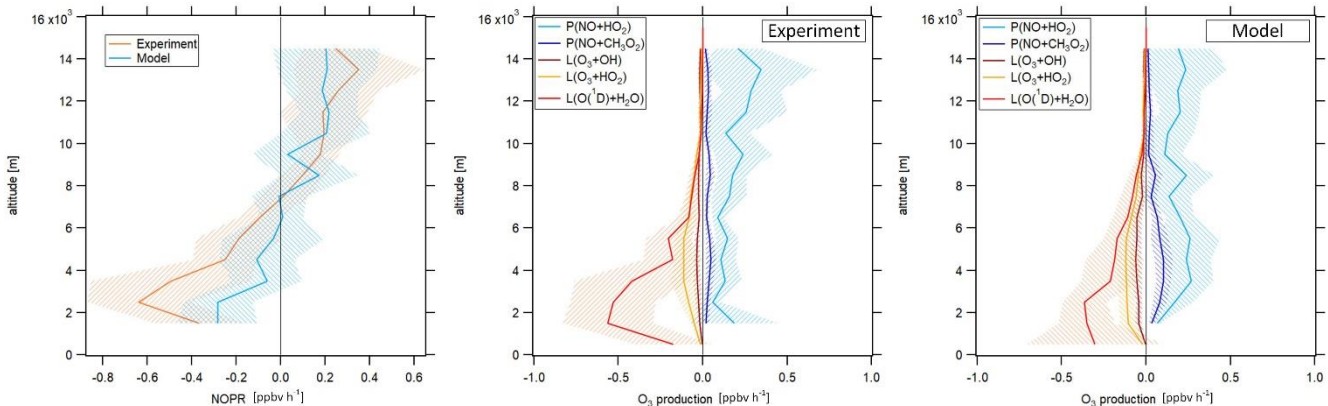

**Figure 5: Vertical profile of tropospheric NOPRs calculated based on Eq. 6 during CAFE-Africa (left graph). The orange and blue lines represent NOPR calculations based on measured and simulated data, respectively. The middle and right graphs show the components of net ozone production in Eq. 6 derived from experimental data (middle graph) and simulated data (right graph). The figures have been filtered for stratospheric measurements by removing data points for which O₃ exceeds 100 ppbv**

During CAFE-Africa NOPRs derived from in situ measurements varied between -1 ppbv h$^{-1}$ to about 0.6 ppbv h$^{-1}$ within $\pm$ 1 standard deviation of the vertical average. We found net ozone destruction for all altitudes below 7-8 km with a minimum of $(-0.6 \pm 0.2)$ ppbv h$^{-1}$ between 2 and 3 km. A general increase of NOPRs with altitude results in net ozone production of 0.2-0.4 ppbv h$^{-1}$ above 9 km altitude with a maximum of $(0.4 \pm 0.3)$ ppbv h$^{-1}$ between 13 and 14 km altitude. The large standard deviation associated with the vertical profile at 13-14 km altitude reflects the large variation in NOPRs along the flight tracks. The vertical NOPR profile derived from in situ data further shows a rather smooth transition from net ozone destruction to net ozone production between 7-8 km altitude, which is in good agreement with the value estimated by Bozem et al. (2017) for the tropical troposphere over the South American rainforest at latitudes of 5 to 10° N. The negative net ozone tendencies observed between 3 and 5 km altitude for the tropical troposphere stand in opposition to positive net ozone tendencies of about 0.1 ppbv h$^{-1}$ (Zanis et al., 2000a) and balance in net ozone tendencies (NOPR = 0) (HOOVER campaign over Europe; Bozem et al., 2017) deduced from previous measurements at similar altitudes at mid-latitudes.

In general, the vertical tendencies in NOPRs derived from the observations are well reproduced by the NOPR calculation based on simulated data. However, the model calculations indicate a minimum in net ozone destruction at $(-0.3 \pm 0.2)$ ppbv h$^{-1}$ between 1 to 3 km, which represents about half of that derived from the in situ measurements for these altitudes. This underestimation of the measurement by the model is directly related to an underestimation of simulated humidity and $j$(O$^1$D), which are both underestimated by EMAC by about 15-20 % below 4 km altitude (see the right graph of supplement Figure S7 for a comparison of the vertical profiles of measured and simulated $j$(O$^1$D)). The model reports net ozone production of 0.2 ppbv h$^{-1}$ above 8 km (except between 9 and 10 km altitude) and also suggests that the transition from net destruction to net production occurs between 6 and 8 km altitude, which again agrees with the measurement-based calculation. Nevertheless, the atmospheric variability of the (simulation-based) average NOPR profile reveals that transition from net ozone destruction to net ozone production occurs within a wider altitudinal range of 4 to 10 km altitude.

The reaction of NO with $HO_2$ dominates gross ozone production (middle graphic, Figure 5) for NOPRs derived from measured in situ data. Whilst the reaction of NO with $CH_3O_2$ contributes about 0.03 ppbv h$^{-1}$ to gross ozone production throughout the whole troposphere, the vertical average of the ozone production rate from the reaction of NO with $HO_2$ yields 0.1 ppbv h$^{-1}$ at the lowest altitudes with a linear increase ($r^2 \approx 0.6$) to about 0.3 ppbv h$^{-1}$ at 14-15 km altitude. From our observations it follows that the ozone production rate due to the reaction of NO with $HO_2$ is from a factor of 2-3 (below 3 km altitude) to a factor of 10 (above 12 km altitude) stronger than gross ozone production due to the reaction of NO with $CH_3O_2$. For the measurement-based estimate, photolysis of ozone dominates gross ozone loss below 6 km altitude. Between 1 and 2 km, it is largest in absolute values at -0.8 ppbv h$^{-1}$, where it contributes to about 80 % of total gross ozone loss. With increasing altitude, the gross ozone loss rate due to photolysis sharply decreases in absolute value to less than -0.05 ppbv h$^{-1}$ above 8 km altitude. Between 6 and 10 km altitude, (total) gross ozone loss is of the order of -0.15 ppbv h$^{-1}$, mainly due to photolysis of $O_3$ while reaction of $HO_2$ with $O_3$ (-0.05 ppbv h$^{-1}$) and the reaction of OH with $O_3$ (-0.03 ppbv h$^{-1}$) are significantly smaller in absolute values. Above 10 km, gross ozone loss rate decreases to -0.03 to -0.05 ppbv h$^{-1}$. This is mainly due to a diminishing ozone loss via photolysis and reaction with $H_2O$ at low humidity, leaving ozone loss by the reaction of $O_3$ with OH and $HO_2$ at -0.01 to 0.02 ppbv h$^{-1}$ as major loss processes. Ozone loss rates observed above 10 km during CAFE-Africa have only little impact on NOPRs as they balance only about 10-20 % of the absolute value of concurrent gross ozone production rates at these altitudes.

The model generally reproduces the NOPR tendencies in gross production and loss as shown above, yielding net ozone destruction at a rate of –0.1 to –0.3 ppbv h$^{-1}$ below 4 km altitude, which is significantly lower in absolute values than the measurement based calculation. This is due to a combination of a weaker loss term due to photolysis and of a larger production term due to the reaction of NO with $HO_2$ and $CH_3O_2$ represented in the model. The contribution of the reaction of NO with $HO_2$ represents a vertically constant value of about 0.2-0.3 ppbv h$^{-1}$ with and a slightly larger production rate from the reaction of NO with $CH_3O_2$ at 0.05 to 0.1 ppbv h$^{-1}$ than inferred from the measurements. Above 10 km altitude, EMAC reproduces the relative and absolute tendencies of the particular gross ozone loss rates remarkably well.

Our results are comparable to a previous study on NOPRs derived from in situ airborne observations at similar latitudes over the rainforest in South America (Bozem et al., 2017). Bozem et al. (2017) report net ozone destruction of -0.2 to -0.6 ppbv h$^{-1}$ between 2 to 4 km and net ozone production between 7 and 9 km altitude with the transition from net ozone destruction to net ozone production occurring at 7 km, similar to our results. Below 6 km altitude gross ozone loss is dominated by photolysis in both studies (Bozem et al., 2017). Bozem et al. (2017) found net ozone production in the continental PBL layer. In this study $O_3$ destruction prevails, most likely due to the absence of large emission sources in the proximity of the Cape Verde Islands. In the marine boundary layer both studies tend towards net ozone destruction (Bozem et al., 2017).

Our results add to the understanding of photochemical net ozone production in the upper troposphere of the region. Using a photochemical trajectory model initiated by in situ measurements, Real et al. (2010) derived photochemical net ozone production rates of 2.6 ppbv/day over a period of 10 days downwind of West Africa. Our study supports the findings by Real et al. (2010) by underlining that photochemical ozone production in the upper troposphere over the tropics is positive at about

0.2-0.4 ppbv h$^{-1}$, which supports the concept of significant photochemical ozone production in the upper troposphere of the region. Note that during CAFE-Africa, measurements at low altitudes were generally performed over the Atlantic Ocean. Hence, we cannot compare to previous results from Saunois et al. (2009) reporting ozone production ranging from 0.25 to 0.75 ppbv h$^{-1}$ in the continental boundary layer over West Africa.

In the following, we investigate the spatial distribution of NOPRs derived from measured and simulated data. Figure 6 shows the color-coded spatial, tropospheric distribution of upper tropospheric (> 12 km altitude) NOPRs calculated from observations (left plot) and model simulated data (middle plot). The right plot shows the spatial, average tropospheric distribution of the point-by-point NOPR(model)/NOPR(measurement) ratio. Note that a few single data points exceed the given color scales. Also note that NOPR calculations based on observational data are restricted to periods of simultaneous availability of a number

measured species and parameters so that data gaps will be more likely than for spatial distributions of in situ NO or O$_3$.

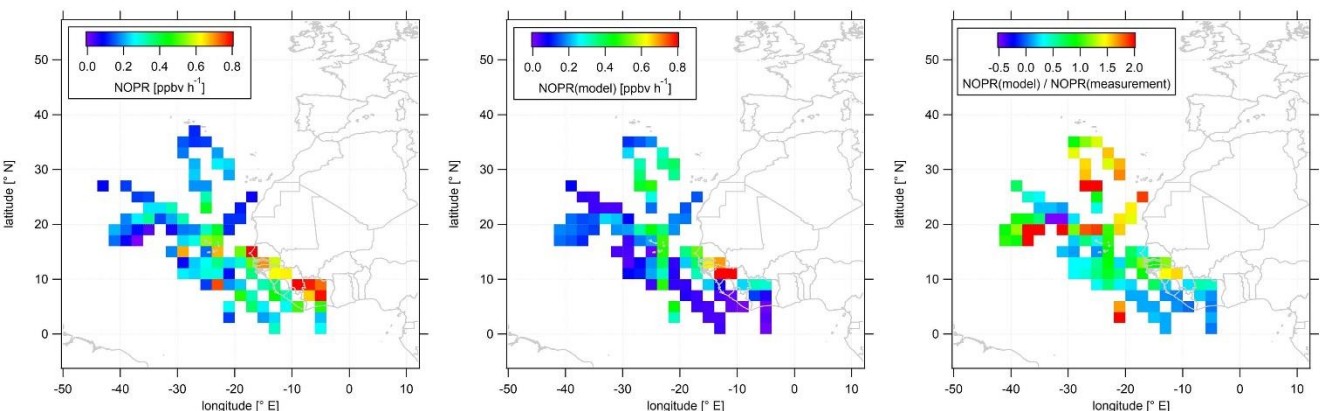

**Figure 6: Color-coded spatial, tropospheric distributions of calculated NOPRs above 12 km during CAFE-Africa. The left and middle graphs show the spatial distribution of measured and simulated O$_3$, respectively. The right graph shows the spatial distribution of the NOPR(model)/NOPR(measurement) ratio. The figures have been filtered for stratospheric measurements by**
**removing data points for which O$_3$ exceeds 100 ppbv.**

The spatial distribution of NOPRs calculated based on measured data shows the already discussed, generally positive net ozone production tendencies in the upper tropical troposphere, but with distinct, characteristic regional features. While NOPRs are generally of the order of $(0.2 \pm 0.1)$ ppbv h$^{-1}$ north of 16° N and west of -20° E, spatially averaged NOPRs in the area of the ITCZ are ~0.4 ppbv h$^{-1}$ at several locations. Largest spatially averaged NOPRs based on the observations (> 0.8 ppbv h$^{-1}$) are

found over tropical West Africa, mirroring strong NO enhancements (see Figure 3). Nevertheless, the highest NOPR values are inferred over the Ivory Coast although NO is lower than over Guinea or over Senegal, where the NOPR calculation yields comparable, but slightly smaller values. Over the Ivory Coast ozone formation is mainly driven by large HO$_2$ mixing ratios of up to 15-20 pptv (see supplement Figure S9 for spatial distributions of in situ measured and simulated OH and HO$_2$ data). Similar as for NO, the spatial distribution further suggest a longitudinal increase of NOPRs towards the West African coast

reflecting the general absence of LNO$_x$ over oceanic areas and increased lightning flash rates over the tropical parts of West and Central Africa (Williams and Satori, 2004; Collier and Hughes, 2011).

The spatial distribution of NOPRs calculated based on simulated data largely follows the spatial distribution of simulated NO. Although the model indicates lowest NOPRs of less than 0.2 ppbv h$^{-1}$ over most oceanic area, NOPRs derived from simulated data exhibit values of about 0.4 ppbv h$^{-1}$ at several locations between -20 and -30° E, which correlate with enhancements in

NO retrieved from EMAC. Over Africa, EMAC yields significant enhancements in NOPRs only between 10 and 14° N over Senegal, where NO is also enhanced.

The strong dependence of ozone formation on ambient NO concentrations for both measurement and model raises the question to which extent ozone formation was NO$_x$-limited. Figure 7 shows NOPRs calculated based on measured and simulated data in orange and blue, respectively, aggregated to a bin width of 0.025 ppbv of NO on the $x$-axis. There is only limited data

coverage for measured NO above 0.325 ppbv and for simulated NO above 0.15 ppbv. See supplementary Table ST3 for the number of data points in each NO mixing ratio bin.

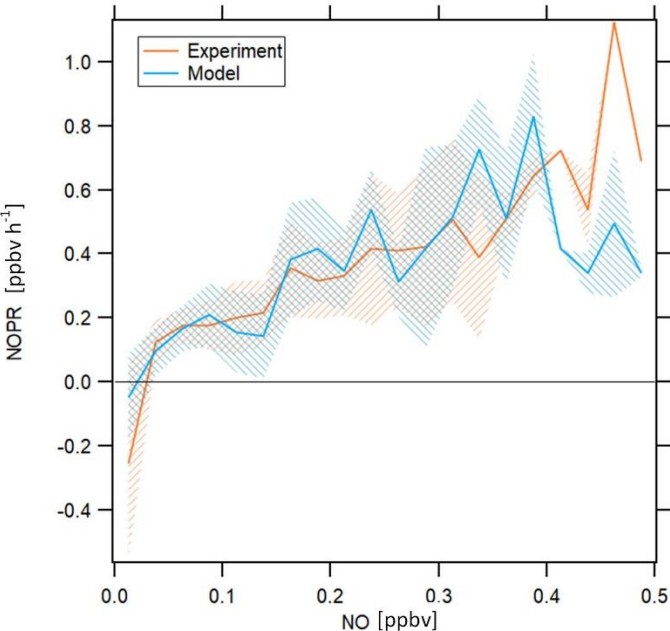

**Figure 7: NOPRs derived from measured and simulated data as a function of NO mixing ratios. The orange and blue lines represent average profiles calculated based on measured and simulated data, respectively. The blue and orange shading represent ±1 standard**
**deviations of the average lines. The profiles have been filtered for stratospheric measurements by removing data points for which O$_3$ exceeds 100 ppbv.**

NOPRs derived from both observations and model simulations exhibit similar dependencies on ambient NO mixing ratios. Both measurements and model simulations show net ozone destruction below 30 and 20 pptv, respectively, and a nearly linear increase in NOPRs with increasing ambient NO mixing ratios above this threshold value, with an NOPR increase of 0.1 to

0.15 ppbv h$^{-1}$ per 100 pptv increase in NO. Especially the NO compensation mixing ratio (for which ozone production equals ozone loss) reproduces results from previous studies remarkably well. Cantrell et al. (2003) report NO compensation mixing ratios between 10 and 30 pptv over the Pacific, depending on whether modelled or measured HO$_2$ and RO$_2$ is used. A study conducted by Zanis et al. (2000b) for the Swiss Alps also reports balance in ozone production for similar NO compensation mixing ratios. Due to low data coverage above 0.4 ppbv we cannot resolve with much certainty whether beyond this value the

increase in NOPR will continue. Note that one possible limitation of this figure arises from the fact that the data aggregated in the respective NO mixing ratio bins stem from different atmospheric layers and origins, which causes the spiky signature of the profile for both measurement and model. However, both model simulations and observations indicate that O$_3$ in the upper troposphere in the tropics is NO$_x$-limited.

**4 Conclusion**

We presented in situ observations of NO, O$_3$ and a number of species involved in photochemical O$_3$ formation obtained in the upper tropical troposphere above the Atlantic Ocean and West Africa and compared these experimental results to simulated data retrieved from the global EMAC chemistry-climate model. Our results corroborate the overall eightfold increase of lightning flash rates over land compared to oceanic areas, and the associated NO production (Christian et al., 2003), as well as the notion that tropical Africa is one of the world's lightning hot spots (Williams and Satori, 2004) where large amounts of

NO are naturally produced in the process of convection. Observed NO mixing ratios reveal a typical vertical average profile with lowest NO mixing ratios of less than 20 pptv in the free and middle troposphere and highest mixing ratios of 150-200 pptv above 12 km altitude. We report highest NO (> 0.4 ppbv) in the latitudinal range of the ITCZ (5° N to 15° N) and moreover over tropical West Africa. While we find overall good agreement when comparing average profiles of observed and EMAC model simulated NO, large deviations are sometimes found for point-to-point comparisons. The model does not reproduce the

largest NO enhancements over West Africa and instead predicts highest NO values above 12 km altitude over large areas of the North Atlantic, which highlights the importance of an accurate representation of lightning NO in the model. Based on in situ measurements we found 60-70 ppbv O$_3$ in the upper tropical troposphere, which is well reproduced by the model. While the average vertical profile of NOPRs derived from in situ measurements varied vertically between -0.6 ppbv h$^{-1}$ between 2 and 4 km altitude and 0.2-0.4 ppbv h$^{-1}$ in the upper tropical troposphere, with a crossover in O$_3$ formation at around 8 km. A

spatial distribution of NOPRs in the upper tropical troposphere created based on experimental in situ data indicates highest values over the West African continent, which is a result of large NO and HO$_2$ over the particular regions. Although the model simulations largely reproduce the observation-based NOPR values, this is at least partly due to compensating effects, e.g. low NO in the model is partly associated with enhanced HO$_2$ leading to locally increased NOPRs in the simulations. Overall both the observations and the model simulations exhibit a nearly linear dependency of NOPRs on ambient NO indicating NO$_x$-

limitation of O$_3$-formation.

**Data availability**

Data used in this study are available to all scientists agreeing to the CAFE-Africa data protocol at https://doi.org/10.5281/zenodo.4442616.

**Author contributions**

IT, CMN, JL and HF designed the study. IT wrote the manuscript. IT and CMN processed and analyzed the data. IT and UP performed the NO, CO and $CH_4$ measurements during the campaign. IT processed the NO, CO and $CH_4$ data. DM, HH, MM and RR performed the OH and $HO_2$ measurements. BB supervised measurements and processed the actinic flux data. MZ supervised measurements and processed the water vapor data. FO supervised measurements and processed the $O_3$ data. AP generated model data.

**Competing interest**

The authors declare that they have no conflict of interest.

**Acknowledgements**

We acknowledge the collaborations with Forschungszentrum Jülich, Karlsruhe Institute of Technology, Heidelberg University, Deutsches Zentrum für Luft- und Raumfahrt and Wuppertal University during the CAFE-Africa campaign. We thank all involved in the CAFE-Africa project for a successful campaign.

**Appendix: Acronyms and abbreviations**

**General**

| | |
|---|---|
| CAFE-Africa | **C**hemistry of the **A**tmosphere: **F**ield **E**xperiment in **Africa** |
| HALO | **H**igh **A**ltitude and **Lo**ng-range research aircraft |

**Scientific**

| | |
|---|---|
| CLD | **C**hemi**l**uminescence **d**etector |
| ECHAM5 | Fifth generation **E**uropean **C**entre **Ham**burg general circulation model |
| EMAC | **E**CHAM/**M**ESSy **A**tmospheric **C**hemistry **m**odel |
| FAIRO | **F**ast **Air**borne **O**zone Instrument |
| HORUS | **H**ydr**O**xyl **R**adical measurement **U**nit based on fluorescence **S**pectroscopy instrument |
| $HO_x$ | OH + $HO_2$ |
| ITCZ | **I**nter-**T**ropical **c**onvergence **z**one |
| LIF | **L**aser **i**nduced **f**luorescence |
| MBL | **M**arine **b**oundary **l**ayer |
| MESSy | **M**odular **E**arth **S**ubmodel **Sy**stem |
| MF | **M**easurement **f**light |
| NOPR | **N**et **o**zone **p**roduction **r**ate |
| $NO_x$ | NO + $NO_2$ |
| PBL | **P**lanetary **B**oundary **L**ayer |
| PSS | **P**hotostationary **s**teady **s**tate |
| QCLAS | **Q**uantum **c**ascade **l**aser **a**bsorption **s**pectroscopy |
| SHARC | **S**ophisticated **H**ygrometer for **A**tmospheric **R**esear**C**h) |
| SLM | **S**tandard **l**itre per **m**inute |

| 555 | TDL | **T**unable **d**iode **l**aser |
| | TMU | **T**otal **m**easurement **u**ncertainty |
| | T/O | **T**ake **o**ff |
| | VOC | **V**olatile **or**ganic **c**ompound |
| | UTC | **C**oordinated **u**niversal **t**ime |
| 560 | UTLS | **U**pper **t**roposphere/**l**ower **s**tratosphere |

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
