# Peer review of "Central role of nitric oxide in ozone production in the upper tropical troposphere over the Atlantic Ocean and West Africa"

_Atmospheric Chemistry and Physics, 2021_

## Referee Comment (RC2)

**Central role of nitric oxide in ozone production in the upper tropical troposphere over the Atlantic Ocean and West Africa, I. Tadic et al., ACPD, 2021**

**General Description:**

The authors use aircraft observations of ozone and NO to estimate net ozone production rates in the upper troposphere over the Atlantic Ocean and West Africa. They determine that ozone production is $NO_x$-limited. My recommendation is substantial changes to the manuscript before this should be considered for publication in ACP. These are detailed below.

**General Comments:**

The introduction does not provide the reader with context for what's known about the region and the time period sampled. The introduction details well-established ozone chemistry, rather than referring to prior publications that focus on ozone chemistry in the region. These include, but are not be limited to, publications that make use of observations from previous and ongoing flight campaigns in West Africa and over the Atlantic (AMMA, DACCIWA, MOZAIC, IAGOS, CARABIC, ATom). URLs to some relevant publications are provided in the References section of this review. An introduction specific to the target region would clarify whether the finding in this study is in dispute, known already, or contrary to what's been found before.

Similarly, the results section should be compared to results published in the literature from previously published research.

The measurement techniques the researchers use are established techniques, so instead dedicate the methods section to describing the aspects of these that are unique and relevant to your study.

Avoid unfamiliar and unnecessary acronyms when there are mathematical symbols or familiar letters that would be easier for the reader to follow. NOPR, RMU, and TMU are not universally familiar. For NOPR, for example, consider rather using $P(O_3)$net. $P(O_3)$gross could then be used to distinguish the two. Many acronyms in the paragraph starting on line 111 seem unnecessary and make it a challenge to read.

**Specific Comments:**

Line 118: How does the flight ceiling compare to the tropopause height during the flight campaign?

Figure 1 caption: Provide dates instead of flight numbers. The former are more meaningful.

Line 190: What domain does the model simulate? Global?

Section 2.4: Are soil $NO_x$ emissions in the model? If so, give the name and reference of the inventory. If not, would this contribute to the model-measurement discrepancy in NO in the lower troposphere?

Line 198: What is "tg"? Do you mean "Tg"?

Line 211: Format of the in-text citation of Tadic et al. is incorrect.

Line 213: What does "Therewith" mean? Is it necessary?

Lines 217-221: This is a very lengthy sentence that makes for challenging reading.

Lines 219: Tell the reader what "PSS" is.

Lines 225-226: The dominant loss pathway for HCHO leading to the formation of $HO_2$ is photolysis. If this loss pathway is also taking into account, is $HO_2$ production from HCHO still negligible?

Equation (6): Is the righthand side of this equation just "$P(O_3)gross – L(O_3)$"? If so, consider rewriting it to this so that it is clearer that this relates to what's given in equations (2) and (5).

Line 231: It's not clear whether the quoted values are calculated in this work or are from a previous study. Reword for clarity.

Figure 2: Considering the simulated NO variability extends into the negative, provide the min and max values of the modelled and observed NO.

Lines 252-253: Why would there be influence from the stratosphere if this has been removed using an ozone concentration threshold?

Line 309: Avoid subjective words like "Interestingly".

Line 433: Change "Please note that there …" to "There …".

Line 432: Was there any doubt or contention that the upper troposphere is $NO_x$-limited over this region?

Figure 7: Give the number of data points in each NO concentration bin.

Line 445: Should "tropical troposphere" be "upper troposphere in the tropics", as this is the focus of the study according to the title.

**References:**

A sample of a few publications relevant to ozone formation in West Africa. There are more.

https://acp.copernicus.org/articles/9/6135/2009/
https://acp.copernicus.org/articles/20/10611/2020/acp-20-10611-2020.pdf
https://science.sciencemag.org/content/304/5676/1483
https://acp.copernicus.org/articles/19/3257/2019/acp-19-3257-2019.pdf
https://hal.archives-ouvertes.fr/hal-00569542/document
https://acp.copernicus.org/articles/7/1193/2007/acp-7-1193-2007.pdf

---

## Author Response (AR1)

**Reply on Referee Comment RC1 on acp-2021-52**

*In the following the comments of the referee are presented (in black) alongside with our replies (in blue) and changes made to the manuscript (in red).*

General statement: The manuscript presents original and valuable experimental results accompanied by global model calculations. Furthermore, it is generally well written and presented. I suggest acceptance of the manuscript for publication, but I have a few minor comments to be considered before the final acceptance.

Dear reviewer, thank you very much for reviewing our manuscript and for the insightful comments. Below we provide detailed responses to your comments.

**Comment 1**: line 35: The wording "NOx is a toxic gas" sounds rather odd as NOx is not a single gas. To avoid misunderstandings the wording could be revised.

We have revised the sentence. It now says in line 35: Both NO and $NO_2$ are toxic gases which degrade surface air quality and regulate the abundance of secondary tropospheric oxidants.

**Comment 2**: line 41: The sentence needs revision.

We have revised the sentence. It now says: The U.S. Clean Air Act identified ozone as a criteria air pollutant in the 1970s (Jaffe et al., 2018).

**Comment 3**: line 84: You may also add some earlier references on NOx-VOC sensitivity of ozone production by Sillman.

We have added the already cited study of Sillman et al. "Photochemistry of ozone formation in Atlanta, GA – models and measurements" (1995) as a reference and another study by Sillman et al., "$O_3$-$NO_x$-VOC sensitivity and $NO_x$-VOC indicators in Paris: Results from models and Atmospheric Pollution Over the Paris Area (ESQUIF) measurements" to the list of references and as a reference to line 84: (Sillman et al., 1995; Sillman et al., 2003; Duncan et al., 2010, Nussbaumer and Cohen, 2020, Tadic et al., 2020).

**Comment 4**: lines 85-86: Please add a reference for the lifetime of NOx.

We have added a reference to lines 85-86 (Beirle et al., 2010).

**Comment 5**: There are a number of NOPR studies based on in situ HOx or ROx measurements by aircraft or at high altitudes stations which could be considered, e.g. Cantrell et al., 1996, Zanis et al., 2000, Cantrell et al., (2003a), Ren et al., (2008) Olson et al., (2012).

We have added Zanis et al. (2000a; The Role of In Situ Photochemistry in the Control of Ozone during Spring at the Jungfraujoch (3,580 m asl) – Comparison of Model Results with Measurements; https://doi.org/10.1023/A:1006349926926) and Cantrell et al. (2003; Peroxy radical behavior during the Transport and Chemical Evolution over the Pacific (TRACE-P) campaign as measured aboard the NASA P-3B aircraft; https://doi.org/10.1029/2003JD003674) to the list of references.

We have added the following sentence to the manuscript in line 367 (referencing Zanis et al., 2000a): The negative net ozone tendencies observed between 3 and 5 km altitude for the tropical troposphere stand in opposition to positive net ozone tendencies of about 0.1 ppbv h$^{-1}$ (Zanis et al., 2000a) and balance in net ozone tendencies (NOPR $\approx$ 0) (HOOVER campaign over Europe; Bozem et al., 2017) deduced from previous measurements at similar altitudes at mid-latitudes.

We have added the following sentence to the manuscript in line 449 (referencing Cantrell et al., 2003): Especially the NO compensation mixing ratio (for which ozone production equals ozone loss) reproduces results from previous studies remarkably well. Cantrell et al. (2003) report NO compensation mixing ratios between 10 and 30 pptv over the Pacific, depending on whether modelled or measured $HO_2$ and $RO_2$ is used. A study conducted by Zanis et al. (2000b) for the Swiss Alps also reports balance in ozone production for similar NO compensation mixing ratios. Note that this second amendment addresses **Comment 17** of our reply. The respective reference (Zanis et al., 2000b) has been added to the list of references.

**Comment 6**: line 211: Should rather be "is practically one or unit" instead of unity.
We have revised line 211 accordingly.

**Comment 7**: line 211: Should be Tadic et al. (2017).
We have revised the passage accordingly.

**Comment 8**: line 235: You may also add some earlier references for the calculation of net ozone production (e.g. Lin et al., 1986).
We have added the suggested reference to the list of references and to line 235/236. However it should be Lin et al., 1988 (https://doi.org/10.1029/JD093iD12p15879) instead of 1986. We have further added Cantrell et al. (2003) as a reference for this sentence.

**Comment 9**: line 242-243: You may add a reference for the selection of the 100 ppbv criterion for stratospheric ozone. For example, see Prather et al., 2011. Other model intercomparison studies generally utilized a chemical tropopause defined at the 150 ppbv.
We have added Prather et al. (2011) to the list of references. Line 242ff now says (the underlined passage is new): Data are filtered for stratospheric influence by removing all data points for which concurrent $O_3$ is larger than 100 ppbv; a conservative criterion which has been discussed by Prather et al. (2011).

**Comment 10**: lines 251-253: The attribution of high NOx above 12 km to lightning NOx rather than NOx rich stratospheric air is rather speculative, unless if there are some indications from the model results of references for that. Mind also the simultaneous relatively smooth increase of both NO and O3 (as you also mention in page 10) which may point influence of stratospheric air.
Our argumentation is based the fact that the tropopause is located at about 16-18 km altitude at the ITCZ, which is still about 3-5 km above typical (highest) cruising altitudes of 12 – 14 km during the campaign. Second, although both NO and O3 show a slight increase above 12 km, the vertical CO profile shows only a slight decrease in average mixing ratios from about 100 ppbv around 12 km altitude to 80 ppbv around 15 km altitude which is statistically insignificant within $\pm 1$ standard deviation of the vertical average CO mixing ratio. Assuming that stratospheric influence did play a (more) dominant role in terms of high NOx, the decrease in CO should be stronger than observed.
Also we have created two additional figures (added at the end of this reply) showing 2-D latitudinal/altitudinal distributions of measured, tropospheric NO and O3 during the campaign. Especially the latitudinal/altitudinal NO distribution shows rather local enhancements at the latitudinal range of the ITCZ than intrusion from the stratosphere at the subtropical jet streams.
We have further added these two figures discussed here to the supplement (as supplement Figure S5 and Figure S6) and redefined the numbering of the following supplement Figures accordingly. A short

passage has been added to the manuscript in line 343: Furthermore, supplementary Figures S5 and S6 show 2-D latitudinal-altitudinal distributions of measured, tropospheric NO and $O_3$, respectively.

**Comment 11**: line 264: At around 6 km it seems that there is an ozone layer of possible stratospheric origin. You may check this with relevant model diagnostics (e.g. specific humidity, potential vorticity or O3S if it is available from the simulation).
There is generally a lower data coverage for altitudes in the free and middle troposphere (between 4 and 10 km). The increase in modelled $O_3$ at 6 km altitude arises from a few data points with increased mixing ratios at this altitude. A vertical profile of modelled humidity does not reproduce stratospheric influence.

**Comment 12**: lines 280-281: This does not necessarily mean that you totally exclude the influence of mixing with air of stratospheric origin.
No, we do not exclude mixing with air of stratospheric origin. To clearify this, we have added a short notice after the respective sentence in line 281f (underlined passage is new): We again remove stratospheric measurement data by only considering those for which $O_3$ was below 100 ppbv. Note that this does not necessarily exclude influence of mixing with air of stratospheric origin.

**Comment 13**: line 308: "…is shown.." should be deleted.
Thanks for noticing. We have removed "is shown" from the sentence.

**Comment 14**: line 368: Although the effect of humidity can be implied from factor $\alpha$ of Eq. 4 maybe it is also interesting adding in the supplementary material the observed and simulated specific humidity values.
We agree that is makes sense to add a comparison of the vertical profiles of observed and simulated humidity. We have added the following underlined sentence in line 349: We provide a vertical profile of $\alpha$ calculated based on Eq. 4, for which we obtain good agreement between measurements and simulations, for which we refer to the left graph of Figure S7 in the supplement. Supplementary Figure S7 also provides a comparison of vertical profiles of measured and simulated $H_2O$ mixing ratios. The respective comparison of measured and modelled H2O mixing ratios has been added to supplement Figure S7 (which already shows the intercomparison of the vertical profile of $\alpha$ and $j(O1D)$). The revised Figure S7 (updated in the supplement) is included at the end of our reply.

**Comment 15**: line 379: Should rather be: "… is from a factor of 2-3 (below 3 km altitude) to a factor of 10 (above 12 km altitude) stronger …"
We have applied the suggested change.

**Comment 16**: Figure 7 is interesting showing the NO dependence of NORP as well as the ozone compensation point (the NO level at which NORP is roughly zero). One possibly limitation is the fact that the aggregated bins correspond to different atmospheric layers with different atmospheric characteristics which can possibly induce the spiky signal in the figure.
We agree that the spiky signature in the profile could be due to the variety of different air masses measured during the campaign and corresponding to a certain bin. We have added the following sentence to the passage below the figure (line 453ff): Note that one possible limitation of this figure arises from the fact that the data aggregated in the respective NO mixing ratio bins stem from different atmospheric layers and origins, which causes the spiky signature of the profile for both measurement and model.

**Comment 17**: line 441: You may also take into consideration the ozone compensation point which was derived in previous studies in the free troposphere and which agrees well with these values (see e.q. Zanis et al., JGR, 2000)

Zanis, P., Monks, P. S., Schuepbach, E., Carpenter, L. J., Green, T. J., Mills, G. P., Bauguitte, S., and Penkett, S. A.: In situ ozone production under free tropospheric conditions during FREETEX '98 in the Swiss Alps, J. Geophys. Res., 105, D19, https://doi.org/10.1029/2000JD900229, 2000b has been added to the list of references. **Comment 17** is being addressed within the answer to **comment 7**.

[Figure]

**Figure S5: Latitudinal/altitudinal distribution of measured, tropospheric NO obtained during the campaign. The data have been aggregated and averaged over a grid width of 2 degree latitude and 1 km altitude.**

[Figure]

**Figure S6: Latitudinal/altitudinal distribution of measured, tropospheric O₃ obtained during the campaign. The data have been aggregated and averaged over a grid width of 2 degree latitude and 1 km altitude.**

[Figure]

**Figure S7: Vertical, tropospheric profile of $\alpha$ calculated based on measured and simulated data during CAFE-Africa (left graph). Vertical, tropospheric profile of H₂O mixing ratios calculated based on measured and simulated data during CAFE-Africa (middle graph). Vertical, tropospheric profile of $j(O^1D)$ (measured and simulated) obtained during CAFE-Africa (right graph). The orange and blue traces represent measured and simulated results, respectively.**

**Reply on Referee Comment RC2 on acp-2021-52**

*In the following the comments of the referee are presented (in black) alongside with our replies (in blue) and changes made to the manuscript (in red).*

**General description**: The authors use aircraft observations of ozone and NO to estimate net ozone production rates in the upper troposphere over the Atlantic Ocean and West Africa. They determine that ozone production is NOx-limited. My recommendation is substantial changes to the manuscript before this should be considered for publication in ACP. These are detailed below.

Dear reviewer, thank you very much for reviewing our manuscript and for the insightful comments. Below we provide detailed responses to your comments.

**General comments**: The introduction does not provide the reader with context for what's known about the region and the time period sampled. The introduction details well-established ozone chemistry, rather than referring to prior publications that focus on ozone chemistry in the region. These include, but are not be limited to, publications that make use of observations from previous and ongoing flight campaigns in West Africa and over the Atlantic (AMMA, DACCIWA, MOZAIC, IAGOS, CARIBIC, ATom). URLs to some relevant publications are provided in the References section of this review. An introduction specific to the target region would clarify whether the finding in this study is in dispute, known already, or contrary to what's been found before.

Similarly, the results section should be compared to results published in the literature from previously published research.

We agree that it makes sense to provide the reader with information on ozone chemistry in the region by including observations from previous and ongoing flight campaigns. We have added a new paragraph to line 101ff (prior to the sentence "In the present study we characterize the distribution of NO and …):

A number of previous studies have performed measurements in the region of interest, the troposphere over the Atlantic Ocean and the West Africa (Lelieveld et al., 2004; Aghedo et al., 2007; Saunois et al., 2009; Real et al., 2010; Bourgeois et al., 2020). Lelieveld et al. (2004) indicated that positive ozone trends in the marine boundary layer over the Atlantic are likely caused by an increase in anthropogenic emissions of nitrogen oxides. Aghedo et al. (2007) showed that lightning acts as a major source of tropospheric $NO_x$, leading to a significant increase in middle and upper tropospheric ozone over the African continent. Saunois et al. (2009) described results from airborne measurements in the region during the AMMA project. Deploying a two-dimensional model for further analysis, Saunois et al. determined positive trends in photochemical net ozone production in the boundary layer over West Africa. There are also results from the ATom airborne mission, which measured vertical profiles of $O_3$ in the troposphere over the Atlantic Ocean (Bourgeois et al., 2020), which we will use to validate the results presented here. Real et al. (2010) investigated downwind $O_3$ production in pollution plumes in the mid and upper troposphere and determined mean net ozone production rates of 2.6 ppbv/day over a period of 10 days. However, studies reporting on vertical profiles and spatial distributions of nitric oxide, ozone and net ozone production rates as part of one coherent measurement project in the troposphere over the West African continent and the Atlantic Ocean are absent.

We added the here mentioned five studies to the list of references:

Aghedo, A. M., Schultz, M. G., and Rast, S.: The influence of African air pollution on regional and global tropospheric ozone, *Atmos. Chem. Phys.*, **7**, 1193-1212, https://doi.org/ 10.5194/acp-7-1193-2007, 2007.

Bourgeois, I., Peischl, J., Thompson, C. R., Aikin, K. C., Campos, T., Clark, H., Commane, R., Daube, B., Diskin, G. W., Elkins, J. W., Gao, R.-S., Gaudel, A., Hintsa, E. J., Johnson, B. J., Kivi, R., McKain, K., Moore, F. L., Parrish, D. D., Querel, R., Ray, E., Sánchez, R., Sweeney, C., Tarasick, D. W., Thompson, A. M., Thouret, V., Witte, J. C., Wofsy, S. C., and Ryerson, T. B.: Global-scale distribution of ozone in the remote troposphere from the ATom and HIPPO airborne field missions, *Atmos. Chem. Phys.*, **20**, 10611–10635, https://doi.org/10.5194/acp-20-10611-2020, 2020.

Lelieveld, J., van Aardenne, J., Fischer, H., de Reus, M., Williams, J., and Winkler, P.: Increasing Ozone over the Atlantic Ocean, *Science*, **304**, Issue 5676, 1483-1487, https://doi.org/10.1126/science.1096777, 2004.

Real, E., Orlandi, E., Law, K. S., Fierli, F., Josset, D., Cairo, F., Schlager, H., Borrmann, S., Kunkel, D., Volk, C. M., McQuaid, J. B., Stewart, D. J., Lee, J., Lewis, A. C., Hopkins, J. R., Ravegnani, F., Ulanovski, A., and Liousse, C.: Cross-hemispheric transport of central African biomass burning pollutants: implications for downwind ozone production, *Atmos. Chem. Phys.*, **10**, 3027–3046, https://doi.org/10.5194/acp-10-3027-2010, 2010.

Saunois, M., Reeves, C. E., Mari, C. H., Murphy, J. G., Stewart, D. J., Mills, G. P., Oram, D. E., and Purvis, R. M.: Factors controlling the distribution of ozone in the West African lower troposphere during the AMMA (African Monsoon Multidisciplinary Analysis) wet season campaign, *Atmos. Chem. Phys.*, **9**, 6135-6155, https://doi.org/10.5194/acp-9-6135-2009, 2009.

A discussion of the $O_3$ vertical profile with respect to results from the ATom mission has been included in the manuscript in line 300ff:

$O_3$ profiles observed in this study are in good agreement with results from the ATom mission (Bourgeois et al., 2020). For the June-August season, Bourgeois et al. show that in the tropical troposphere $O_3$ increased with altitude to 50 ppbv at 5-6 km whereas above 9 km $O_3$ varied from 40 to 80 ppbv, supporting the results presented here (see Figures 9 and 10 in Bourgeois et al., 2020).

Another section has been added to discuss the relevance of photochemical ozone production derived in this study in line 437:

Our results add to the understanding of photochemical net ozone production in the upper troposphere of the region. Using a photochemical trajectory model initiated by in situ measurements, Real et al. (2010) derived photochemical net ozone production rates of 2.6 ppbv/day over a period of 10 days downwind of West Africa. Our study supports the findings by Real et al. (2010) by underlining that photochemical ozone production in the upper troposphere over the tropics is positive at about 0.2-0.4 ppbv h$^{-1}$, which supports the concept of significant photochemical ozone production in the upper troposphere of the region. Note that during CAFE-Africa, measurements at low altitudes were generally performed over the Atlantic Ocean. Hence we cannot compare to previous results from Saunois et al. (2009) reporting ozone production ranging from 0.25 to 0.75 ppbv h$^{-1}$ in the continental boundary layer over West Africa (2009).

The measurement techniques the researchers use are established techniques, so instead dedicate the methods section to describing the aspects of these that are unique and relevant to your study.

We consider the section describing the measurement methods and techniques to be essential to the study. We have added a short sentence in line 197 emphasizing the applicability of the instrumental set up for observations on an airborne platform:

All instruments deployed on the aircraft have been developed to meet the high standards of airborne measurements in terms of operability, accuracy and sensitivity.

Avoid unfamiliar and unnecessary acronyms when there are mathematical symbols or familiar letters that would be easier for the reader to follow. NOPR, RMU, and TMU are not universally familiar. For NOPR, for example, consider rather using P(O3)net. P(O3)gross could then be used to distinguish the two. Many acronyms in the paragraph starting on line 111 seem unnecessary and make it a challenge to read.

We agree that it makes sense to unify "total measurement uncertainty" (TMU) and "relative measurement uncertainty" (RMU), which describe in principle the same phenomenon. We have therefore replaced "relative measurement uncertainty"/"RMU" in Table 1 and in the caption of Table 1 by "total measurement uncertainty"/"TMU".

However, in terms of the acronym NOPR we have to state that the acronym/symbol for net ozone production rate" is not unified in the community. Previous studies (Thornton et al., 2002) used the acrnonym "$P(O_3)$" as the gross rate of $O_3$ production, which is equivalent to what is given in our study. Also Real et al. (2010, https://doi.org/10.5194/acp-10-3027-2010) abbreviated "net ozone production" by "$NPO_3$", similar to the acronym "NOPR" mentioned in our study. Last but not least, "NOPR" has already been established in previous studies (Bozem et al., 2017; Tadic et al., 2020). We would like stay with the used acronym "NOPR".

Nevertheless to make it easier to follow the manuscript, we have added short notices in line 231: Note that other studies use P(O₃)gross as an acronym for P(O₃) in Eq. 2.

… and in line 238:

Note that other studies use P(O3)net as an acronym for NOPR in Eq. 6.

**Comment 1**: Line 118: How does the flight ceiling compare to the tropopause height during the flight campaign?

The tropopause is located at about 16-18 km altitude at the ITCZ, which is 3-5 km above the flight ceiling. At higher latitudes (~ 30° N), the tropopause is located at about 13-15 km.

**Comment 2**: Figure 1 caption: Provide dates instead of flight numbers. The former are more meaningful.

We have replaced the flight numbers by flight dates. A revised version of Figure 1 (with revised caption) can be found at the end of this document. We have revised lines 128ff in the manuscript (underlined words are new) to facilitate the attribution of flight dates to flight numbers: For the analysis of the MFs, we consecutively numerate each MF. We start with MF03 on August 07, 2018 for the ferry flight from Oberpfaffenhofen (Germany, Deutsches Zentrum für Luft- und Raumfahrt) to Sal (Cape Verde Islands) and ending with MF16 on September 07, 2018 for the back ferry flight from Sal to Oberpfaffenhofen.

**Comment 3**: What does the model simulate? Global?

Yes, EMAC is a global model. We have clarified this at the beginning of chapter 2.4 by adding the attribute "global" to the description of EMAC (underlined word is new): EMAC is a 3-D global general

circulation, atmospheric chemistry-climate model, which has been used and described in a number of previous studies (Roeckner et al., 2006; Jöckel et al., 2016; Sander et al., 2019; Tadic et al., 2020).

**Comment 4**: Are soil NOx emissions in the model? If so, give the name and reference of the inventory. If not, would this contribute to the model-measurement discrepancy in NO in the lower troposphere?
NOx soil emissions are included in the model. However, there is no inventory itself in the model as NOx soil emissions are calculated online by an algorithm described by Yienger and Levy II (Yienger and Levy II, 1995; Kerkweg et al., 2006). Based on the Yienger and Levy II algorithm, Pozzer et al. have determined a yearly emission of NOx at 6.77 to 7 TgN/yr (see supplement of Pozzer et al., 2007). Weng et al. found that averaged over the last 37 years, global total soil NOx emissions amount to 9.5 TgN/yr (2020), which means that NOx soil emission in the EMAC model might be on the lower end of the most recent estimates. We have added the following sentence to the description of the model in line 216: The $NO_x$ soil biogenic emission flux is calculated based on a semi-empirical emission algorithm implementation by Yienger and Levy II (1995; Kerkweg et al., 2006).
Yienger and Levy II (1995) has been added to references of the manuscript. Kerkweg et al. is already cited in the manuscript.

Yienger, J. and Levy II, H.: Empirical model of global soil-biogenic $NO_x$ emissions, *J. Geophys. Res.*, **100**, 11 447–11 464, https://doi.org/10.1029/95JD00370, 1995.

Pozzer et al.: Simulating organic species with the global atmospheric chemistry general circulation model ECHAM5/MESSy1: a comparison of model results with observations, *Atmos. Chem. Phys.*, **7**, 2527–2550, 2007.

Weng et al.: Global high-resolution emissions of soil $NO_x$, sea salt aerosols, and biogenic volatile organic compounds, *Sci Data*, **7**, 148 (2020).

**Comment 5**: Line 198: What is "tg"? Do you mean "Tg"?
Yes, we mean "Tg". We have revised the unit accordingly (now line 214)

**Comment 6**: Line 211: Format of the in-text citation of Tadic et al. is incorrect.
We have revised the passage accordingly.

**Comment 7**: Line 213: What does "Therewith" mean? Is it necessary?
We agree that "therewith" is redundant. We have removed "therewith" from the sentence.

**Comment 8**: Line 217-221: This is a very lengthy sentence that makes for challenging reading.
We agree that it makes sense to rewrite this sentence. We have also clarified the acronym PSS in line 219 (which addresses comment 9). Line 236ff now say: We further assume photostationary steady state (PSS) for the probed air masses. As the typical time to acquire PSS during CAFE-Africa varied between 40 s at 2 km altitude and about 70-80 s at 15 km altitude (Mannschreck et al., 2004; Tadic et al., 2020), we can calculate the concentration of $CH_3O_2$ by the equation derived by Bozem et al. (2017).

**Comment 9**: Line 219: Tell the reader what "PSS" is.
See answer to comment 8.

**Comment 10**: Lines 225-226: The dominant loss pathway for HCHO leading to the formation of HO2 is photolysis. If this loss pathway is also taken into account, is HO2 production from HCHO still negligible? In the manuscript we have conservatively estimated ambient HCHO mixing ratios at 100 pptv, which represents an upper limit for HCHO and is most likely a factor of 2 too high. Taking into account the photolysis of HCHO (being an additional pathway of HO2 production) results in HO2 production from the reaction of CO with OH being on average a factor of 5 larger than the sum of HO2 production from the sum of the reaction of H2 with OH, the reaction of HCHO with OH and photolysis of HCHO. In combination with the fact that 100 pptv already represents an upper limit for HCHO, we still can assume that the reaction of CO with OH dominates HO2 production during most of the campaign time to estimate CH3O2. Nevertheless, we have incorporated this new finding into the manuscript in line 241ff: Note that the reaction of CO with OH represents the dominant term in HO2 production during CAFE-Africa. Assuming mixing ratios of 500 ppbv and 100 pptv for H2 and HCHO, respectively, we find that HO2 production rate from the reaction of OH with CO is on average 5 times greater than the sum of the HO2 production rates from photolysis of HCHO and the reactions of HCHO and H2 with OH during CAFE-Africa. Note that the assumed mixing ratio of 100 pptv represents a rather conservative upper estimate for HCHO in the upper troposphere.

**Comment 11**: Equation (6): Is the righthand side of this equation just "P(O₃)gross – L(O₃)"? If so, consider rewriting it to this so that it is clearer that this relates to what's given in equations (2) and (5). Yes, the righthand side of this equation is just "P(O₃)gross – L(O₃)". Equation 6 now reads as:

$$\mathrm{NOPR} = P(O_3) - L(O_3) =$$
$$[\mathrm{NO}] \cdot \left( k_{\mathrm{NO+HO_2}}[\mathrm{HO_2}] + k_{\mathrm{NO+CH_3O_2}}[\mathrm{CH_3O_2}] \right) - [O_3] \cdot \left( \alpha \cdot j(O^1D) + k_{\mathrm{OH+O_3}}[\mathrm{OH}] + k_{\mathrm{HO_2+O_3}}[\mathrm{HO_2}] \right)$$

**Comment 12**: Lines 231: It's not clear whether the quoted values are calculated in this work or are from a previous study. Rewrite for clarity. These values represent results from a previous study (Bozem et al., 2017). We have clarified this by revising line 251: In the troposphere, $\alpha$ ranges from about 15 % in the PBL to 1 % in the upper troposphere, where absolute humidity is very low (Bozem et al., 2017).

**Comment 13**: Figure 2: Considering the simulated NO variability extends into the negative, provide the min and max values of the modelled and observed NO. The simulated NO variability extends into the negative due to the large variability of NO data at highest altitudes. Note that the model itself does not yield any negative data at all. However having a few simulated data points with large NO mixing ratios results in a relatively large standard deviation of the respective average and, consequently, in the variability ($\pm$ 1 standard deviation) being negative above 12 km altitude. The min and max mixing ratios of modelled NO are zero and 2.13 ppbv, respectively. The mix and max mixing ratios of observed NO are zero and 0.95 ppbv. This finding has been incorporated into the study in line 284: The minimum and maximum mixing ratios of modelled NO are zero and 2.13 ppbv, respectively. The minimum and maximum mixing ratios of observed NO are zero and 0.95 ppbv, respectively.

**Comment 14**: Lines 252-253: Why would there be influence from the stratosphere if this has been removed using an ozone concentration threshold?

The ozone mixing ratio threshold of 100 ppbv only removes measurements obtained directly in the stratosphere where O3 significantly exceeds 100 ppbv. However, this threshold does not remove all periods with stratospheric influence. Take for instance an (upper) tropospheric air mass with intrinsically low $O_3$. Influence from the stratosphere could result in an increase in $O_3$ which however would not necessarily result in $O_3$ mixing ratios larger than 100 ppbv.

**Comment 15**: Line 309: Avoid subjective words like "interestingly".
We have removed "interestingly" from this sentence.

**Comment 16**: Line 433: Change "Please note that there …" to "There …".
We have revised line 474 accordingly. It now says:  There is only limited data coverage for measured NO above 0.325 ppbv and for simulated NO above 0.15 ppbv. See supplementary table ST3 for the number of data points in each NO mixing ratio bin.
The second sentence of this amendment already addresses comment 18 (to add the number of data points in each NO concentration bin.).

**Comment 17**: Line 432: Was there any doubt or contention that the upper troposphere is $NO_x$-limited over this region?
There was no doubt or contention that the upper troposphere might or might not be $NO_x$-limited. This is a fundamental finding, which arises from the dependency of net ozone production on NO (presented in Figure 7), which has, as far as we know, not been presented before for the upper troposphere over the Atlantic Ocean and West Africa.

**Comment 18**: Give the number of data points in each NO concentration bin.
The number of data points in each NO concentration bin are given in the following table, which has been added to the supplements of this study. We have further revised sentence in Line 474ff: There is only limited data coverage for measured NO above 0.325 ppbv and for simulated NO above 0.15 ppbv. See supplementary table ST3 for the number of data points in each NO mixing ratio bin.

**Table ST3: Number of data points in each NO mixing ratio bin. Note that the data coverage of the simulation is larger than that of the measurement due to gaps in the observational data.**

| NO mixing ratio bin | measurement | simulation |
| --- | --- | --- |
| 0 pptv ≤ NO < 25 pptv | 140 | 226 |
| 25 pptv ≤ NO < 50 pptv | 54 | 70 |
| 50 pptv ≤ NO < 75 pptv | 36 | 40 |
| 75 pptv ≤ NO < 100 pptv | 23 | 60 |
| 100 pptv ≤ NO < 125 pptv | 50 | 38 |
| 125 pptv ≤ NO < 150 pptv | 44 | 24 |
| 150 pptv ≤ NO < 175 pptv | 28 | 7 |
| 175 pptv ≤ NO < 200 pptv | 30 | 7 |

| | | |
|---|---|---|
| 200 pptv ≤ NO < 225 pptv | 21 | 3 |
| 225 pptv ≤ NO < 250 pptv | 15 | 7 |
| 250 pptv ≤ NO < 275 pptv | 11 | 3 |
| 275 pptv ≤ NO < 300 pptv | 12 | 5 |
| 300 pptv ≤ NO < 325 pptv | 11 | 5 |
| 325 pptv ≤ NO < 350 pptv | 3 | 3 |
| 350 pptv ≤ NO < 375 pptv | 2 | 6 |
| 375 pptv ≤ NO < 400 pptv | 3 | 2 |
| 400 pptv ≤ NO < 425 pptv | 1 | 1 |
| 425 pptv ≤ NO < 450 pptv | 5 | 2 |
| 450 pptv ≤ NO < 475 pptv | 1 | 3 |
| 475 pptv ≤ NO < 500 pptv | 1 | 3 |

**Comment 19**: Lines 445: Should "tropical troposphere" be "upper troposphere in the tropics", as this is the focus of the study according to the title.

We have revised lines 445 and 446 accordingly. It now says in line 492ff: However, both model simulations and observations indicate that $O_3$ production in the upper troposphere in the tropics is $NO_x$-limited.

[Figure]

**Figure 1: Spatial orientation of the measurement flight tracks during CAFE-Africa. Note that MF07 (August 17, 2018), MF08 (August 19, 2018) and MF11 (August 26, 2018) had identical flight tracks.**

**Further changes made to the manuscript**

We have made the following additional changes to the manuscript, which are not related to the referee's comments and listed below:

**Change 1:** We have replaced "TDLAS" by "TDL" for the measurement technique/method of $H_2O$ in Table 1 as it is also mentioned as "TDL" in the manuscript.

[revised manuscript text omitted]

Figure S8: Latitudinal profile of NOPRs derived from measured (upper plot) and simulated (lower plot) data above 12 km at a latitudinal resolution of 2 degree. The black and grey traces represent the average and median latitudinal profile, respectively. The green background represents $\pm 1$ standard deviation of the average profile. The red box in the background indicates the supposable location of the ITCZ during August and September. The profiles were filtered for stratospheric measurements by removing data for which concurrent $O_3$ is > 100 ppbv.

[Figure]

**Figure S9: Color-coded spatial, tropospheric distributions of measured and simulated OH and HO₂ above 12 km during CAFE-Africa. The upper graphics show the measurements, the lower graphics show the simulations. Note that the figures are filtered for stratospheric measurements by removing data points for which concurrent O₃ is > 100 ppbv.**

800

**Table ST1: Overview of the scientific measurement flights performed during CAFE-Africa. EDMO is the international civil aviation organization (ICAO) code of the airport of the DLR facility at South Germany, GVAC the ICAO code of the airport on Sal and DGAA the ICAO code of the airport of Accra (Ghana).**

| Flight number | Date of the flight | Purpose/objective of the flight | Flight route |
|---|---|---|---|
| MF03 | 07 August 2018 | ferry flight to Sal (Cape Verde Islands) | EDMO → GVAC |
| MF04 | 10 August 2018 | flight south / biomass burning plume | GVAC → GVAC |
| MF05 | 12 August 2018 | flight south / aged biomass burning plume | GVAC → GVAC |
| MF06 | 15 August 2018 | flight north / stratospheric influence | GVAC → GVAC |
| MF07 | 17 August 2018 | stack flight #1 | GVAC → GVAC |
| MF08 | 19 August 2018 | stack flight #2 | GVAC → GVAC |
| MF09 | 22 August 2018 | flight over West Africa | GVAC → DGAA |
| MF10 | 24 August 2018 | flight northwest / aged biomass burning plume | GVAC → GVAC |
| MF11 | 26 August 2018 | stack flight #3 | GVAC → GVAC |
| MF12 | 29 August 2018 | flight over the ITCZ | GVAC → GVAC |
| MF13 | 31 August 2018 | flight over the ITCZ | GVAC → GVAC |
| MF14 | 02 September 2018 | convective outflow of hurricane "Florence" | GVAC → GVAC |
| MF15 | 04 September 2018 | flight over West Africa | GVAC → GVAC |
| MF16 | 07 September 2018 | ferry flight to Oberpfaffenhofen (Germany) | GVAC → EDMO |

**Table ST2: List of peroxy radicals (with less than four carbon atoms) which were used to estimate RO$_2$ (in analogy to Tadic et al., 2020).**

| Species |
|---|
| HO$_2$ |
| CH$_3$O$_2$ |
| C$_2$H$_5$O2 |
| C$_2$H$_5$CO$_3$ |
| CH$_3$CO$_3$ |
| C3DIALO2 (C$_3$H$_3$O$_4$) |
| CH$_3$CHOHO$_2$ |
| CH$_3$COCH$_2$O$_2$ |
| CH$_3$COCO$_3$ |
| CHOCOCH$_2$O$_2$ |
| CO$_2$H$_3$CO$_3$ |
| HCOCH$_2$CO$_3$ |
| HCOCH$_2$O$_2$ |
| HCOCO$_3$ |
| HCOCOHCO$_3$ |
| HOC$_2$H$_4$CO$_3$ |
| HOCH$_2$CH$_2$O$_2$ |
| HOCH$_2$CO$_3$ |
| HOCH$_2$COCH$_2$O$_2$ |
| HOCH$_2$O$_2$ |
| CH$_3$CHO$_2$CH$_2$OH |
| IC3H7O2 (isopropylperoxy radical) |
| NC3H7O2 (propylperoxy radical) |
| NCCH$_2$O$_2$ |
| NO$_3$CH$_2$CO$_3$ |
| CH$_3$CHO$_2$CH$_2$ONO$_2$ |

810

**Table ST3: Number of data points in each NO mixing ratio bin. Note that the data coverage of the simulation is larger than that of the measurement due to gaps in the observational data.**

| NO mixing ratio bin | measurement | simulation |
|---|---|---|
| 0 pptv ≤ NO < 25 pptv | 140 | 226 |
| 25 pptv ≤ NO < 50 pptv | 54 | 70 |
| 50 pptv ≤ NO < 75 pptv | 36 | 40 |
| 75 pptv ≤ NO < 100 pptv | 23 | 60 |
| 100 pptv ≤ NO < 125 pptv | 50 | 38 |
| 125 pptv ≤ NO < 150 pptv | 44 | 24 |
| 150 pptv ≤ NO < 175 pptv | 28 | 7 |
| 175 pptv ≤ NO < 200 pptv | 30 | 7 |
| 200 pptv ≤ NO < 225 pptv | 21 | 3 |
| 225 pptv ≤ NO < 250 pptv | 15 | 7 |
| 250 pptv ≤ NO < 275 pptv | 11 | 3 |
| 275 pptv ≤ NO < 300 pptv | 12 | 5 |
| 300 pptv ≤ NO < 325 pptv | 11 | 5 |
| 325 pptv ≤ NO < 350 pptv | 3 | 3 |
| 350 pptv ≤ NO < 375 pptv | 2 | 6 |
| 375 pptv ≤ NO < 400 pptv | 3 | 2 |
| 400 pptv ≤ NO < 425 pptv | 1 | 1 |
| 425 pptv ≤ NO < 450 pptv | 5 | 2 |
| 450 pptv ≤ NO < 475 pptv | 1 | 3 |
| 475 pptv ≤ NO < 500 pptv | 1 | 3 |

815